# A deep proteome and transcriptome abundance atlas of 29 healthy human tissues

Dongxue Wang[1],[†] iD, Basak Eraslan[2],[3],[†], Thomas Wieland[4], Björn Hallström[5], Thomas Hopf[4], Daniel Paul Zolg[1], Jana Zecha[1], Anna Asplund[6], Li-hua Li[1], Chen Meng[1], Martin Frejno[1] iD, Tobias Schmidt[1], Karsten Schnatbaum[7], Mathias Wilhelm[1], Frederik Ponten[6] iD, Mathias Uhlen[5], Julien Gagneur[2],[*] iD, Hannes Hahne[4],[**] iD & Bernhard Kuster[1],[8],[***] iD

## Abstract

Genome-, transcriptome- and proteome-wide measurements provide insights into how biological systems are regulated. However, fundamental aspects relating to which human proteins exist, where they are expressed and in which quantities are not fully understood. Therefore, we generated a quantitative proteome and transcriptome abundance atlas of 29 paired healthy human tissues from the Human Protein Atlas project representing human genes by 18,072 transcripts and 13,640 proteins including 37 without prior protein-level evidence. The analysis revealed that hundreds of proteins, particularly in testis, could not be detected even for highly expressed mRNAs, that few proteins show tissue-specific expression, that strong differences between mRNA and protein quantities within and across tissues exist and that protein expression is often more stable across tissues than that of transcripts. Only 238 of 9,848 amino acid variants found by exome sequencing could be confidently detected at the protein level showing that proteogenomics remains challenging, needs better computational methods and requires rigorous validation. Many uses of this resource can be envisaged including the study of gene/protein expression regulation and biomarker specificity evaluation.

**Keywords** human proteome; human transcriptome; proteogenomics; quantitative mass spectrometry; RNA-Seq
**Subject Categories** Genome-Scale & Integrative Biology; Methods & Resources
**Mol Syst Biol. (2019) 15: e8503**

See also: **B Eraslan et al** (February 2019)

## Introduction

Delineating the factors that govern protein expression and activity in cells is among the most fundamental research topics in biology. Although the number of potential protein-coding genes in the human genome is stabilizing at about 20,000, high-quality evidence for their physical existence has not yet been found for all and intense efforts are ongoing to identify these currently ~13% "missing proteins" (Omenn et al, 2017). While it is also generally accepted that the quantities of proteins vary greatly within and across different cell types, tissues and body fluids (Kim et al, 2014; Wilhelm et al, 2014), this has not been analysed systematically for many human tissues. Furthermore, it is not very clear yet how the many anabolic and catabolic processes are coordinated to give rise to the often vast differences in the levels of proteins. Messenger RNA levels are important determinants for protein abundance (Vogel et al, 2010; Schwanhäusser et al, 2011), and extensive mRNA expression maps of human cell types and tissues have been generated as proxies for estimating protein abundance (GTEx Consortium, 2013; Uhlén et al, 2015; Thul et al, 2017). However, other studies have also highlighted the much higher dynamic range of protein than transcript abundance as well as a rather poor correlation of mRNA and protein levels suggesting that further and possibly diverse regulatory elements play important roles (Schwanhäusser et al, 2011; Liu et al, 2016; Franks et al, 2017). Decades of careful research revealed numerous mRNA elements affecting translation or mRNA stability such as codon usage, start

1 Chair of Proteomics and Bioanalytics, Technische Universität München, Freising, Germany
2 Computational Biology, Department of Informatics, Technical University of Munich, Garching bei München, Germany
3 Department of Biochemistry, Quantitative Biosciences Munich, Gene Center, Ludwig Maximilian Universität, München, Germany
4 OmicScouts GmbH, Freising, Germany
5 Science for Life Laboratory, KTH - Royal Institute of Technology, Stockholm, Sweden
6 Science for Life Laboratory, Department of Immunology, Genetics and Pathology, Uppsala University, Uppsala, Sweden
7 JPT Peptide Technologies GmbH, Berlin, Germany
8 Center for Integrated Protein Science Munich (CIPSM), Munich, Germany
*Corresponding author. Tel: +49 89 289 19411; E-mail: gagneur@in.tum.de
**Corresponding author. Tel: +49 8161 976289 0; Fax: +49 8161 976289 1; E-mail: hannes.hahne@omicscouts.com
***Corresponding author. Tel: +49 8161 71 5696; Fax: +49 8161 71 5931; E-mail: kuster@tum.de
†These authors contributed equally to this work

codon context or secondary structure to name a few. However, most of these studies focussed on single or few genes or single cell types or were performed in model organisms distinct from human systems and often did not cover a lot of proteins. Broader scale analyses have more recently become possible owing to advances in proteome and transcriptome profiling technologies, but these have mostly focussed on a single (disease) tissue or the cell-type resolved analysis of protein expression in single tissues (Zhang *et al*, 2014; Mertins *et al*, 2016). To the best of our knowledge, no broad-scale quantitative and integrative analysis of transcriptomes and proteomes across many healthy human tissues has been performed yet that would enable a comprehensive analysis of factors explaining the experimentally observed differences between mRNA and protein expression. Therefore, the purpose of this study was to generate a resource of molecular profiling data at the mRNA and protein level to facilitate the study of protein expression control and proteogenomics in humans. To this end, we analysed 29 major histologically healthy human tissues from the Human Protein Atlas (HPA) project (Uhlén *et al*, 2015) to provide a comprehensive baseline map of protein expression across the human body. As we show below as well as in Eraslan *et al*, 2019, these data can be used in many ways to explore protein expression and its regulation in humans. To facilitate further research on this fundamentally important topic and the many further uses that can be envisaged, all data are available in ArrayExpress (Kolesnikov *et al*, 2015) and proteomeXchange (Vizcaíno *et al*, 2014).

## Results and Discussion

### Comprehensive transcriptomic and proteomic analysis of 29 human tissues

We analysed 29 histologically healthy tissue specimen representing major human organs by label-free quantitative proteomics and RNA-Seq (Fig 1A; see Appendix Figs S1–S6 for the assessment of data quality). Tissues were collected by the HPA project (Fagerberg *et al*, 2014), and adjacent cryosections were used for paired (allele-specific) transcriptome and proteome analysis. RNA-Seq profiling detected and quantified in total 18,072 protein-coding genes with an average of 12,262 ($\pm$ 1,007 standard deviation, SD) genes per tissue (Fig 1B) when using a cut-off of 1 fragment per kilobase million (FPKM; Uhlén *et al*, 2015). Proteomic profiling by mass spectrometry resulted in the identification and intensity-based absolute quantification (iBAQ; Schwanhäusser *et al*, 2011) of a total of 15,210 protein groups with an average of 11,005 ($\pm$ 680 SD) protein groups per tissue at a false discovery rate (FDR) of < 1% at the protein, peptide and peptide-spectrum match (PSM) level (Fig EV1A). Protein identification was based on 277,698 non-redundant tryptic peptides, representing a total of 13,640 genes and, on average, 10,541 ($\pm$ 512 SD) genes per tissue covering, on average, 86% of the expressed genome in every tissue. While the total number of confidently identified proteins in this study is smaller than that of other (community-based) resources such as ProteomicsDB (Schmidt *et al*, 2018) and neXtProt (Gaudet *et al*, 2017; coverage of 15,721 and 17,470 protein-coding genes, respectively), it provides a highly consistent collection of tissue proteomes including the deepest proteomes to date for many of the tissues analysed. It also provides

protein-level evidence for 37 proteins (represented by at least one unique peptide) that are not yet covered by neXtProt (release 2018-01-17; Table EV1). These proteins were validated by synthetic peptides (see PRIDE submission for mirror spectra). Eighteen of these 37 have antibody staining in the current release of the HPA project and all of them show signal in the same tissue they were detected in by MS. This corroborates the detection of these new proteins by an independent method. Eight of these proteins also meet the guidelines of the Human Proteome Project that require $\geq$ 2 peptides for a new protein each with $\geq$ 9 amino acids in length (Deutsch *et al*, 2016). We note that the HPP guidelines use reasonable but ad hoc criteria which are likely too conservative and therefore likely discriminate against further genuine cases. Comparing spectra of endogenous to synthetic peptides is likely the more objective criterion which is why we added mirror plots of all evaluated cases to PRIDE (Zolg *et al*, 2017). The expression levels of the "new" proteins were about a factor 10 below median (iBAQ at log10 scale, 7.4 versus 8.3) which explains why they may have been missed before. Interestingly, 15 of these proteins were detected in the fallopian tube, an organ that has not yet been extensively profiled by proteomics.

Overall, 13,413 protein-coding genes were detected on both transcript and protein levels, and the detected proteins spanned almost the entire range of mRNA expression again indicating very substantial coverage of the expressed proteome (Fig 1C). However, some proteins could not be detected even for highly expressed mRNAs (i.e. higher than the mean mRNA abundance). About 1/3 of these mRNAs were found in testis (478 of 1,408) and no other tissue contained nearly as many highly expressed mRNAs without protein evidence (Fig EV1B). The "missing" proteins in the testis were statistically significantly enriched for processes related to spermatogenesis by gene ontology analysis (clusterProfiler; $n = 82$ genes; BH-adjusted $P = 8 \times 10^{-14}$). Although the rich expression of mRNAs in testis has been known for a long time and exploited for, e.g., the cloning of many genes from cDNAs, the apparent absence of so many testis proteins with high mRNA expression is surprising. This was not due to, e.g., poor coverage of the testis proteome (11,024 detected protein-coding genes) or other obvious technical factors (such as inefficient extraction of membrane proteins or difficulties with identifying small proteins) that would prevent detection of these proteins. Interestingly, almost 300 of these "missing" proteins have also not been detected by antibodies in testis (according to HPA) and nearly 200 have no ascribed molecular function. The inability to detect these proteins by mass spectrometry or antibodies despite high levels of mRNA poses a number of questions. For example, are these proteins rapidly degraded implying specialized (and perhaps transient) functions in testis or sperm functionality? Are they perhaps stabilized in response to egg fertilization? Proteins missing at the lower end of the mRNA expression range (less than mean mRNA abundance) are overrepresented in G-protein-coupled receptor activity ($n = 173$; BH-adjusted $P = 8.3 \times 10^{-50}$), ion channels ($n = 109$; BH-adjusted $P = 7 \times 10^{-10}$) and cytokine-related biology ($n = 76$; BH-adjusted $P = 6 \times 10^{-9}$). The abundance of these proteins may simply have been below the mass spectrometric detection limit or, as described many times, can be difficult to extract from cells owing to the presence of multi-pass transmembrane domains giving rise to few if any MS-compatible tryptic peptides after digestion.

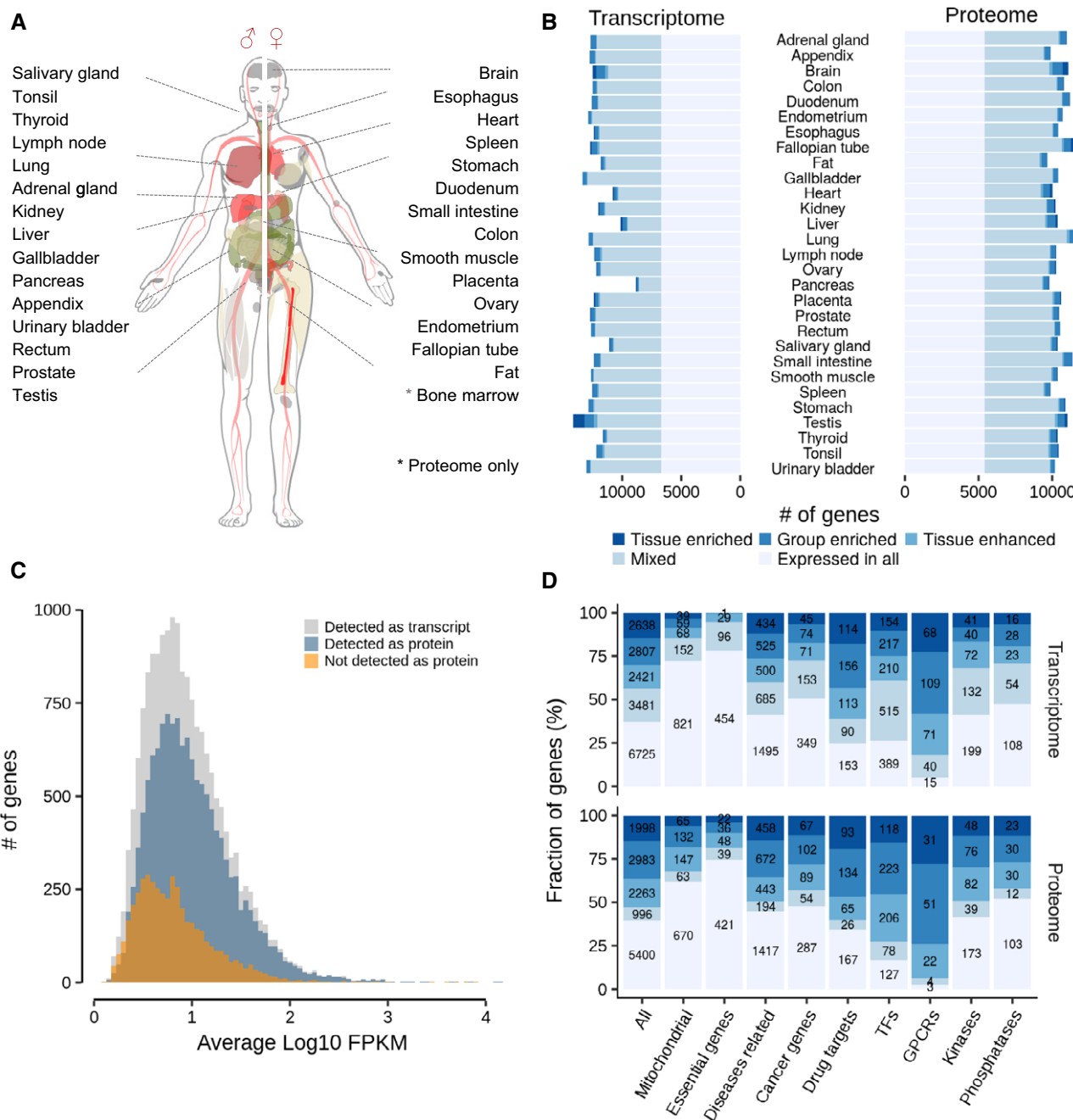

**Figure 1.   Comprehensive proteomic and transcriptomic analysis of 29 human tissues from healthy donors.**

A   Body map of analysed tissues.

B   Number of genes detected on protein and mRNA level in each tissue. The colouring of the bars indicates the fractions of transcripts and proteins that are expressed everywhere or enriched in certain tissues. The full classification is provided in the text.

C   Abundance distribution of all transcripts detected in all tissues (grey); the fraction of detected proteins is shown in blue and the fraction of transcripts for which no protein was detected is shown in orange.

D   Relative distribution and absolute numbers of transcripts and proteins in selected functional classes across the expression categories shown in panel (B). Colours are the same as in panel (B).

To explore which and how many proteins show a tissue-specific expression profile, we applied the classification scheme of Uhlén et al (2015, 2016) previously developed for mRNA profiling and which stratifies genes into the five classes "tissue-enriched" (fivefold above any other tissue), "group enriched" (fivefold above any group of 2–7 tissues), "enhanced" (fivefold above the average of all other tissues), "expressed in all" (expressed in all tissues) as well as "mixed" genes (which do not match the other categories).

Overall, a large fraction of all represented genes was expressed in all tissues: 37% (6,725) at the transcript level and 39% (5,400) at the protein level. However, 43% (7,866) of all transcripts and 53% (7,244) of all proteins showed elevated expression in one or more tissues ("tissue-enriched", "group-enriched" or "tissue-enhanced"). Only 0.73% (on average) of all transcripts and 0.65% of all proteins showed a tissue-enriched profile. Two notable exceptions are brain and testis which exhibit a higher percentage of tissue-enriched proteins and transcripts in line with a recent analysis of RNA-Seq data from the HPA and GTEx projects (GTEx Consortium, 2013). Proteins with more tissue-restricted expression tended to be of slightly lower abundance (Fig EV1C).

For 1,270 of the total 1,998 tissue-enriched proteins detected in our study, antibody staining was available in the HPA. In the 29 tissues that are common between HPA and the current study, 775 proteins were detected in the same tissue lending support to the mass spectrometry-based data presented here. In addition, we compared our tissue-enriched expression data to the targeted MS (PRM) data acquired for about 52 proteins by Edfors *et al* (2016) and 10 tissues that overlapped with our tissue panel (see Appendix Figs S7–S9). Incidentally, the Edfors' study had data on three tissue-enriched proteins. First, myoglobin (MB) was highly tissue-enriched in our data in the heart which was confirmed by the PRM analysis as well as antibody staining in HPA. Second, the protein PDK1 (3-phosphoinositide-dependent protein kinase-1) was also found to be a heart-enriched protein and the PRM data confirmed this. This protein was detected in all tissues by antibody staining but we note that immunohistochemistry (IHC) stains are not quantitative so it is difficult to conclude if broad detection of this protein was due to overstaining or poor antibody specificity. The third example is the protein CANT1 (soluble calcium-activated nucleotidase 1) which we detected as a prostate-enriched protein. Again, this was confirmed by the PRM measurement but was again detected in most tissues by IHC.

The above global trends in transcript and protein tissue expression distributions were also mirrored by functional categories of genes but with some interesting detail (Fig 1D, Table EV4). For example, while the tissue distribution of expression of disease-associated genes followed that of all genes, the expression of drug targets in general and GPCRs in particular was much more tissue-restricted speaking to the notion that proteins may make for better drug targets if they are not ubiquitously expressed (Hao & Tatonetti, 2016). In this context, we point out that our baseline map of protein expression across the human body may be of general value for drug discovery as one can, e.g., quickly examine the expression profile of a particular target of interest, to help better understand adverse clinical effects and off-target mechanisms of action of drugs. For instance, a recent study revealed phenylalanine hydroxylase (PAH) as an off-target of the pan-HDAC inhibitor panobinostat (Becher *et al*, 2016). Our map of protein expression shows that PAH is abundantly expressed in liver (and kidney) which is also the major site of hydroxylation in the human body (Matthews, 2007), indicating that the liver is the major site where panobinostat exerts its detrimental effects, i.e. leading to decreased tyrosine levels, and eventually hypothyroidism in affected patients. In contrast, essential genes (Blomen *et al*, 2015; Hart *et al*, 2015; Wang *et al*, 2015) as well as mitochondrial genes were found in the vast majority of all tissues in line with their central roles for maintaining cellular homeostasis.

Despite the differences in detail, our dataset confirms, at the protein level, that there is a core set of ubiquitously expressed genes/proteins and that individual tissues are not strongly characterized by the categorical presence or absence of mRNAs or proteins but rather by quantitative differences (Geiger *et al*, 2013). This is also evident from an analysis of the most divergently expressed proteins or transcripts that shows enrichment of proteins related to the functional specialization of the respective tissue (Fig EV1D, Table EV3).

### mRNA and protein expression

The relationship between mRNA and protein expression has been studied extensively over the past years and there continues to be debate in terms of how the various correlations that can be computed may be interpreted in terms of technical artefacts or biological meaning (Liu *et al*, 2016; Fortelny *et al*, 2017; Franks *et al*, 2017; Wilhelm *et al*, 2017). While it is beyond the scope of the current study to attempt to reconcile the different views, the extensive data on both mRNA and protein expression provided in this resource should help to eventually bring clarity. Therefore, in the following, we confine our analysis of the expression data to a few basic points we nonetheless deem important.

The dynamic range of transcripts detected by RNA-Seq spanned about four orders of magnitude and that of proteins detected by mass spectrometry spanned eight orders of magnitude (Fig 2A; see Appendix Fig 10 for the corresponding plot using copy numbers that show essentially the same characteristics; Table EV5). This difference alone explains (at least in part) the overall higher coverage of the expressed proteome by RNA-Seq compared to that of LC-MS/MS. This is because there is limited "sequencing capacity" particularly in mass spectrometry. Thus, detecting very low-abundance molecules will be harder, the wider the dynamic range of expression and the lower the sampling depth is. For example, the (paired-end) RNA data provided (on average) 18 M reads per tissue. Those 18 M reads are distributed across 4 orders of magnitude of abundance with an inevitable bias to the higher abundant transcripts. The MS data only provided (on average) ~76,000 peptides and ~284,000 identified tandem mass spectra (peptide to spectrum matches; PSMs) per tissue and these are distributed over eight orders of magnitude also with a bias for the more abundant proteins. As a result, it is currently much easier to cover many genes by RNA-Seq than it is to cover the same number by LC-MS/MS.

As noted before, the much wider dynamic range at the protein level implies that protein synthesis and protein stability play an important role in determining protein levels beyond mRNA levels (Schwanhäusser *et al*, 2011; Vogel & Marcotte, 2012). Similarly, the number of protein molecules produced per molecule of mRNA appears to be much larger for high- than for low-abundance transcripts, leading to a nearly quadratic relationship between mRNA levels and protein levels in every tissue (slope of 2.6 in Fig 2B for brain and between 1.8 and 2.7 for all 29 tissues, Fig EV2A; Appendix Fig S11). While this observation has been made before in yeast (Csárdi *et al*, 2015), this study shows that it is a general phenomenon. The effect may be rationalized by cellular economics such that genes encoding highly abundant proteins not only express high mRNAs levels, but also encode regulatory elements that favour high translation efficiency and high protein stability (Vogel *et al*, 2010). The often vast differences in mRNA and protein expression

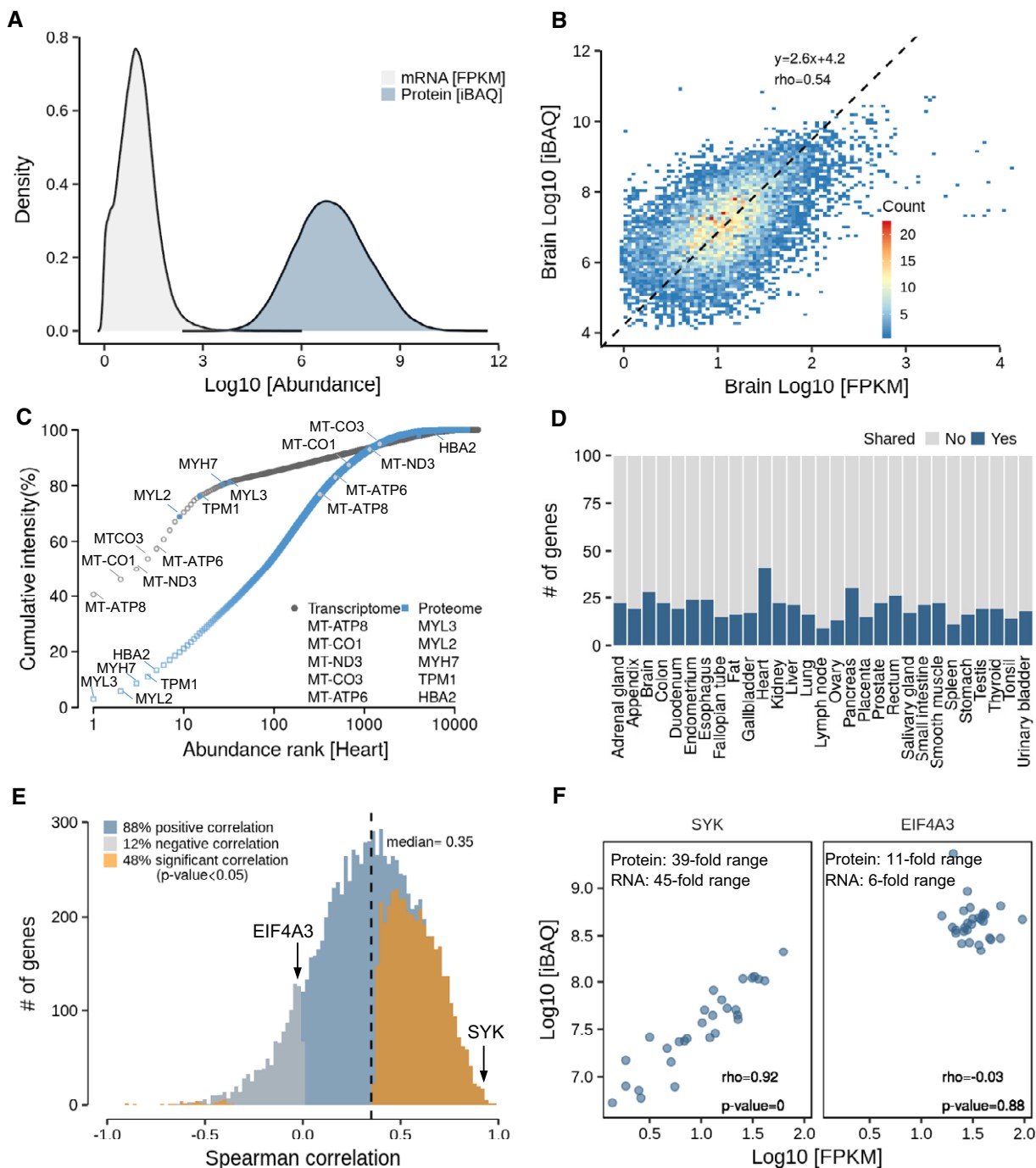

**Figure 2.   Analysis of protein and transcript expression levels within and across tissues.**

A   Distribution of global transcript and protein abundance in all tissues. It is apparent that the dynamic range of protein expression (iBAQ scale) exceeds that of mRNA expression (FPKM scale; see Appendix Fig S10 for the corresponding plot for RNA and protein copy numbers).

B   Protein-to-mRNA abundance plot for brain tissue. The slope of the regression line indicates that high-abundance mRNAs give rise to more protein copies per mRNA than low-abundance mRNAs.

C   Ranked abundance plot of proteins and transcripts in human heart. While the 10 most abundant transcripts cover almost 70% of all transcripts in this tissue, the corresponding proteins only represent about 20% of the total protein.

D   Analysis of the number of genes that are shared among the 100 most abundant transcripts and proteins. Regardless of the tissue, the fraction of shared genes rarely exceeds 20%.

E   Correlation analysis of protein-to-RNA abundance (in log10 scale) across tissues, resulting in almost 90% positive correlations. The proteins highlighted in the next panel are marked.

F   Examples for proteins that show high (SYK, left panel) or no (EIF4A3, right panel) correlation of protein/RNA ratios across tissues. While the former indicates that different tissues express different quantities of SYK, EIF4A3 expression appears to be similar in all tissues.

within a tissue can also be visualized by plotting the ranked order of relative intensities of transcripts and proteins (Fig 2C, Appendix Fig S12). For example, in the heart (an extreme case), 41% of the total mRNA quantity (by FPKM) represents a single protein (MT-ATP8) and nearly 60% of the total mRNA covers just five transcripts (all coding for mitochondrial proteins). In contrast, about 13% of the total protein quantity (by iBAQ) is contributed by five proteins (four of which are myosins and one represents a "contamination" from blood present in the tissue). One would expect the heart to be rich in both protein families owing to the contractile function of the organ which requires a lot of energy. While it is possible that some of the mitochondrial proteins are underrepresented in quantitative terms (because, e.g., MT-ATP8 is a very small protein (7 kDa) and its iBAQ value may therefore not reflect its true quantity or because our lysis conditions may not have solubilized this organelle with high efficiency), it is surprising that even among the 100 most highly expressed mRNAs and proteins, only about 20% are the same (Fig 2D). This overlap only increases to about 60% for the 5,000 most abundant proteins and transcripts (Fig EV2B). The above reflects why RNA–protein abundance plots generally show only modest correlation. The above rank order lists of transcripts and proteins are also quite different between tissues with the spleen showing the opposite characteristics compared to the heart, and the lung showing a more even distribution of transcript and protein levels (Fig EV2C and D).

We find that many proteins are often expressed at broadly similar levels across human tissues (say within a factor 10; Fig EV2E). It is, therefore, not very surprising that the correlation of mRNA/protein ratios across tissues is generally not very strong (Fig 2E; median 0.35). Still, there is positive correlation in ~90% of all cases and almost half are also statistically significant. This distribution is not affected by requiring detection of a protein in 10, 20 or all 29 tissues (see Appendix Figs S13–S15). However, great care has to be taken when interpreting such distributions. As shown in Fig 2F, the transcript and protein levels of the tyrosine kinase SYK span an expression range of 45-fold and 39-fold, respectively (natural scale), and are highly correlated across tissues reflecting the specialized function of the protein in T- and B-cell biology. In contrast, RNA and protein expression of EIF4A3 (a DEAD-box RNA helicase involved in translation initiation) only spanned sixfold and 11-fold between tissues and showed no correlation. We note that cases such as the latter are merely the result of technical variation in the measurement or genuinely similar expression levels in most tissues reflecting the roles of these proteins in central biological processes in all tissues (Appendix Fig S16; Wilhelm *et al*, 2017).

It is noteworthy that proteomes correlate stronger between tissues (median of 0.77) than transcriptomes (median of 0.67; Fig 3A; see Appendix Fig S17). It is possible that, because of the fact that the dynamic range of protein levels is larger than that of RNA, small biological or technical variations of individual genes may or may not have impact on the overall rankings (Fortelny *et al*, 2017; Franks *et al*, 2017). It might, however, also imply that there are (hitherto not very clear) mechanisms in cells that "buffer" the protein quantities against changes in mRNA abundance (Liu *et al*, 2016; Kustatscher *et al*, 2017). The strongest correlations for both transcripts and proteins were found for the anatomically adjacent small intestine and duodenum. At the proteome level, the brain showed clear differences to other proteomes and gastrointestinal

organs appear to be more similar to each other. Visualizing the transcriptome and proteome profiles in a plane using co-inertia analysis (CIA; Culhane *et al*, 2005) indicate that mRNA and protein levels are more similar to each other within tissues than between tissues (Fig 3B) also reflected by an RV coefficient of 0.77 (a multivariate generalization of the squared Pearson correlation coefficient). Moreover, the CIA grouped several tissues according to similarities in their physiological function with tissues of the immune system and of the gastrointestinal tract representing the largest groups. It is interesting to note that this clustering appears to be driven by the cellular composition of individual tissues (Table EV6). For instance, the appendix co-clusters with the spleen, lymph node and tonsil and all four tissues contain a large fraction of lymphocytes (Fig 3C, blue panel). Similarly, the duodenum and small intestine comprise a large proportion of (intestinal) glandular cells, which are important determinants of the molecular make-up of those tissues (Fig 3C, grey panel). All the above illustrates that there must be multiple molecular factors and mechanisms determining the quantitative expression of proteins. This particular aspect of the present mRNA/protein expression resource may be particularly useful for the community as it provides a rich data source for the study of protein expression control (see also Eraslan *et al*, 2019).

### Proteogenomic characterization of human tissues

One aspect of the data we cover in more detail in this study is the considerable interest in the community to use proteomics data to annotate genomes, often referred to as proteogenomics. With matched RNA-Seq and proteomics data at hand, we set out to assess the merits of proteogenomics at several levels. First, we investigated the identification of protein isoforms. Based on RNA-Seq data, it has been suggested that human cell types typically express one dominant isoform (Gonzàlez-Porta *et al*, 2013; Ezkurdia *et al*, 2015). In proteomics, isoforms are often more difficult to distinguish because the identification of proteins is inferred from the underlying peptide data. Given that many proteins contain conserved stretches of amino acids and the fact that the median sequence coverage achieved for each protein is limited (here between 14 and 25%; Fig EV3A), many potential isoforms may not be covered by unique peptides and some peptides may also match to multiple entries in comprehensive sequence collections such as Ensembl (102,450 entries). This often leads to the identification of a so-called protein group rather than one specific protein or isoform thereof. Illustrated by the proteomic data obtained from trypsin-digested tonsil (Fig 4A), only 14% of all protein groups contained one single protein when searching the MS data against Ensembl. However, when searching the same data against a protein sequence database constructed from the tissue-specific RNA-Seq data, the proportion of single entry protein groups increased to 53% (see Appendix Fig S18 for all tissues). In this way, we were able to identify 60,519 non-redundant isoforms by RNA-Seq for 18,072 genes and confirm 15,257 by proteomics for 11,833 genes across the 29 tissues (Fig EV3B, Tables EV2 and EV8). The same analysis also showed that there were rather few proteins that were detected with more than one isoform in the tonsil tissue (see Appendix Fig S19).

One way to improve the detection of isoforms is to increase the sequence coverage in the proteomic data. To this end, we performed an ultra-deep proteomic analysis of tonsil tissue by applying seven

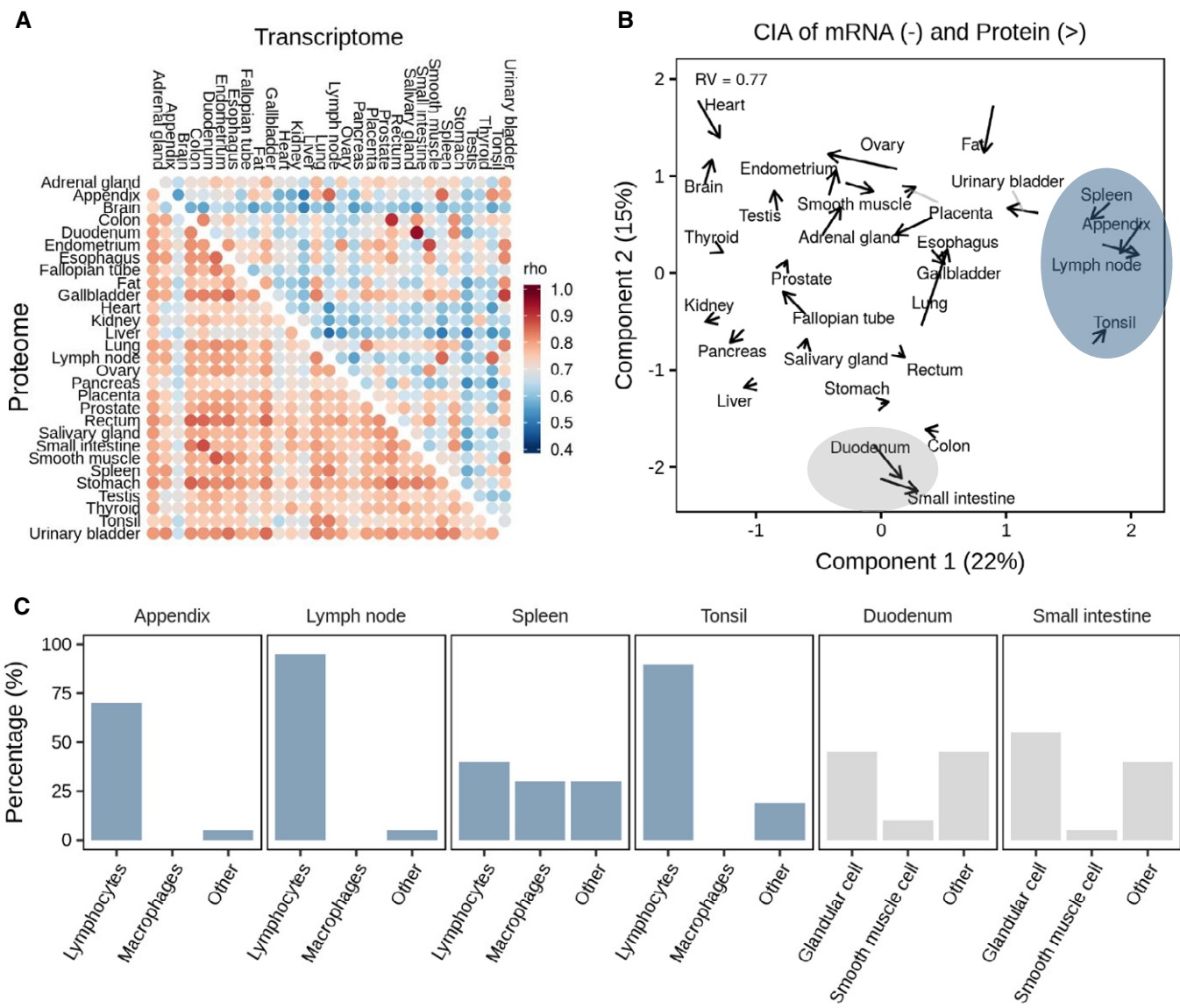

**Figure 3. Correlation analysis of protein and transcript expression levels.**

A Global correlation analysis of proteomes versus proteomes and transcriptomes versus transcriptomes across human tissues. It is apparent that proteomes correlate stronger across tissues than transcriptomes.

B Co-inertia analysis of transcriptome and proteome levels of all 29 tissues (arrow base: transcriptome; arrow head: proteome) showing that the information carried by transcriptomes and proteomes was closer to each other in the same than across different tissues. Grey lines are used to aid identifying tissue names for the respective arrows. Shaded areas highlight tissues that are related by their molecular profiles.

C Average cellular compositions of tissues highlighted in panel (B) showing that the molecular similarities in their transcriptomes and proteomes are driven by similarities in cell types.

different proteases (trypsin, LysC, LysN, GluC, ArgC, AspN and chymotrypsin) and three peptide fragmentation techniques (HCD, CID and EThcD/ETD). This resulted in the identification of 11,569 protein groups (10,288 genes), represented by 421,073 non-redundant peptides leading to a median protein sequence coverage of 54 % (Fig EV3C–E; Table EV7) when searched against Ensembl. Of these protein groups, 2,201 could be unambiguously linked to a single isoform. When searched against protein sequences derived from the tonsil-specific RNA-Seq data, we identified 10,592 protein groups, among which 6,293 represented a particular isoform

identified by unique peptides (Fig 4A; Table EV7). The above shows that isoform calling on the protein level is indeed quite successful particularly when matched RNA-Seq data are available. We note though that because most isoforms were detected by very few isoform-unique peptides, confident quantification of the different isoforms of the same gene found in the same tissue can be difficult and may require targeted MS assays rather than shotgun approaches to be accurate. In this context, it is also worth mentioning that there is no clear consensus in the proteomics and transcriptomics communities as to how quantitative values should be allocated to particular

proteins or transcripts. While it is custom in proteomics to use the parsimonious approach (i.e. allocate all iBAQ intensity to the protein with the highest overall evidence), it is custom to distribute RNA-Seq reads covering shared sequences across multiple transcripts containing that sequence (Trapnell *et al*, 2010). Unfortunately, there is currently no software available that would enable the systematic analysis of using either approach for both types of data. This should be attempted in the future because it would not be surprising if these differences in quantification approaches would add substantially to the poor correlation of mRNA and protein levels (or their ratios). In addition, there is currently no tractable way to determine which allele of a gene gave rise to a detected protein or isoform thereof.

To assess the ability of proteomics to detect genetic variants such as single amino acid variants (SAAV), we generated whole exome sequencing (WES), RNA-Seq and ultra-deep proteomics data for tonsil tissue. In the WES data, the average exon coverage was 98× and 97% of the exons were covered > 20× providing a sound basis for the identification of SAAVs. Variant calling and filtering of WES data resulted in 9,848 high-quality, non-synonymous point mutations (i.e. nonsense and missense variants excluding I > L and L > I variants that cannot be distinguished by mass spectrometry), representing 5,527 human genes and including 6,112 heterozygous and 3,736 homozygous variants (Fig 4B, Table EV8). In the RNA-Seq data, 3,524 of the 9,848 genomic variants (36%; 2,171 heterozygous and 1,353 homozygous cases; representing 2,428 genes) were covered sufficiently (≥ 10×) to assess their genotype. The reason for the substantial loss of coverage in the RNA-Seq versus exome data is because (i) not all genes are expressed from both alleles in a given tissue and (ii) even at a sequencing depth of 18 million reads, the dynamic range of mRNA abundance is too high to cover all transcripts and variants many times over.

It has been noted before that the identification of SAAVs by proteomics is challenging and plagued by false positives in standard database searching regimen because (i) the tandem mass spectra used for database searching are often not of very high quality, (ii) these spectra often do not contain the complete amino acid sequence information of the underlying peptide and (iii) the current FDR statistics for peptide/protein identification do not translate well to variant calling on the peptide level. As a result, random matches can and will frequently occur raising substantial concern about the quality of part of the current proteogenomic literature (Nesvizhskii, 2014). When searching our proteomic data against concatenated sequences obtained from the WES data, Ensembl and UniProt and requiring an identification by both Mascot and Andromeda as well as a number of further criteria (for details, see methods), we identified 1,942 candidate peptides mapping to 724 of the 9,848 (non-canonical) exome variants (7.4% of total; 400 heterozygous and 324 homozygous cases). These peptide variants are all missense mutations (Table EV8). For 41% of the heterozygous cases (165 out of 400), we obtained peptide-level evidence for the canonical and alternative variant, while for the remaining cases, we only identified the alternative variant (235).

For validation, candidate peptide spectra were compared to those of synthetic reference standards (Zolg *et al*, 2017). To this end, we attempted the synthesis of reference peptides for all 724 alternative variants and obtained such peptides for 574 cases. Automated spectral angle analysis (Toprak *et al*, 2014) provided evidence for 238

variants (SA ≥ 0.7, Mascot ion score ≥ 50) including 109 heterozygous and 129 homozygous cases (Fig 4B; Table EV8; see PRIDE for mirror plots). Manual inspection of the above 724 candidate peptides identified 204 unique alternative variant sites of which 158 were also found in the SA analysis. The variants that passed our (conservative) filtering criteria merely represent 2.4% of all variants detected at the exome level, 6.7% of the variants detected at the mRNA level and 32% of the candidates suggested by database searching. When tracing the confidently identified peptide variants back to the proteomic workflow, it became clear that the vast majority of all variants are represented by peptides generated by trypsin, LysC and ArgC cleavage and using the standard HCD fragmentation technique. In addition, the confirmation rate (using the synthetic peptide reference standards) for tryptic peptides was also much higher than that of the other enzymes (Fig 4C; synthetic peptides were not measured by EThcD/ETD). This can be attributed to the fact that trypsin-like peptides generally show well-predictable fragmentation behaviour and that most bioinformatic tools are optimized for use with data generated from tryptic digestion of proteomes.

While the above shows that some of the variants detected on the nucleotide level could be confirmed at the protein level, the overall success rate was low. We note here that this was mainly not due to lack of expression of the underlying gene because the proteomic data cover 76% of all expressed tonsil genes (9,287 of 12,203 mRNA-Seq genes), 48% (2,633 of 5,527) of all the genes for which variants were detected by exome sequencing and 76% (1,836 of 2,428) by RNA-Seq, respectively (further discussed below). Instead, the main reasons for poor coverage of SAAVs at the proteome level are the still limited sensitivity and dynamic range of detection, limited peptide coverage of a protein and the insufficient coverage of amino acids in peptide mass spectra along with shortcomings in peptide identification algorithms. Further reasons include that annotated variants may actually not exist at the protein level (e.g. because of sequencing/calling errors), they may not be translated, they may be rapidly degraded to an extent that they are not detectable at steady state etc. Even for the successful cases, the analysis clearly shows that SAAV detection by proteomics requires very rigorous validation in order to be credible.

Recent research showed that there is more heterogeneity in gene models than previously anticipated, as a result of, e.g., alternative translation initiation sites (aTIS; Na *et al*, 2018) and there is an ongoing debate in the community whether or not long non-coding RNAs (lncRNAs) can be translated into proteins (Chen *et al*, 2017). Ribosomal profiling showed that thousands of potential aTIS may exist and that ~40% of all lncRNAs can at least engage the ribosome (Kearse & Wilusz, 2017). In order to explore if our resource can provide protein evidence for such cases, we used a database search strategy (Marx *et al*, 2017) in which we queried all LC-MS/MS files against combined sequences from (i) a curated lncRNA database (GENCODE v.25), (ii) a database containing protein sequences derived from alternative translation initiation sites (see Materials and Methods), (iii) GENCODE, (iv) UniProt and (v) the tissue-specific RNA-Seq data. Any potential lncRNA or aTIS peptide was required to originate from one single sequence collection only (i.e. lncRNA or aTIS and no other database), be identified by both Mascot and Andromeda, to fulfil stringent score cut-offs (see Materials and Methods), to be backed up by the expression of the

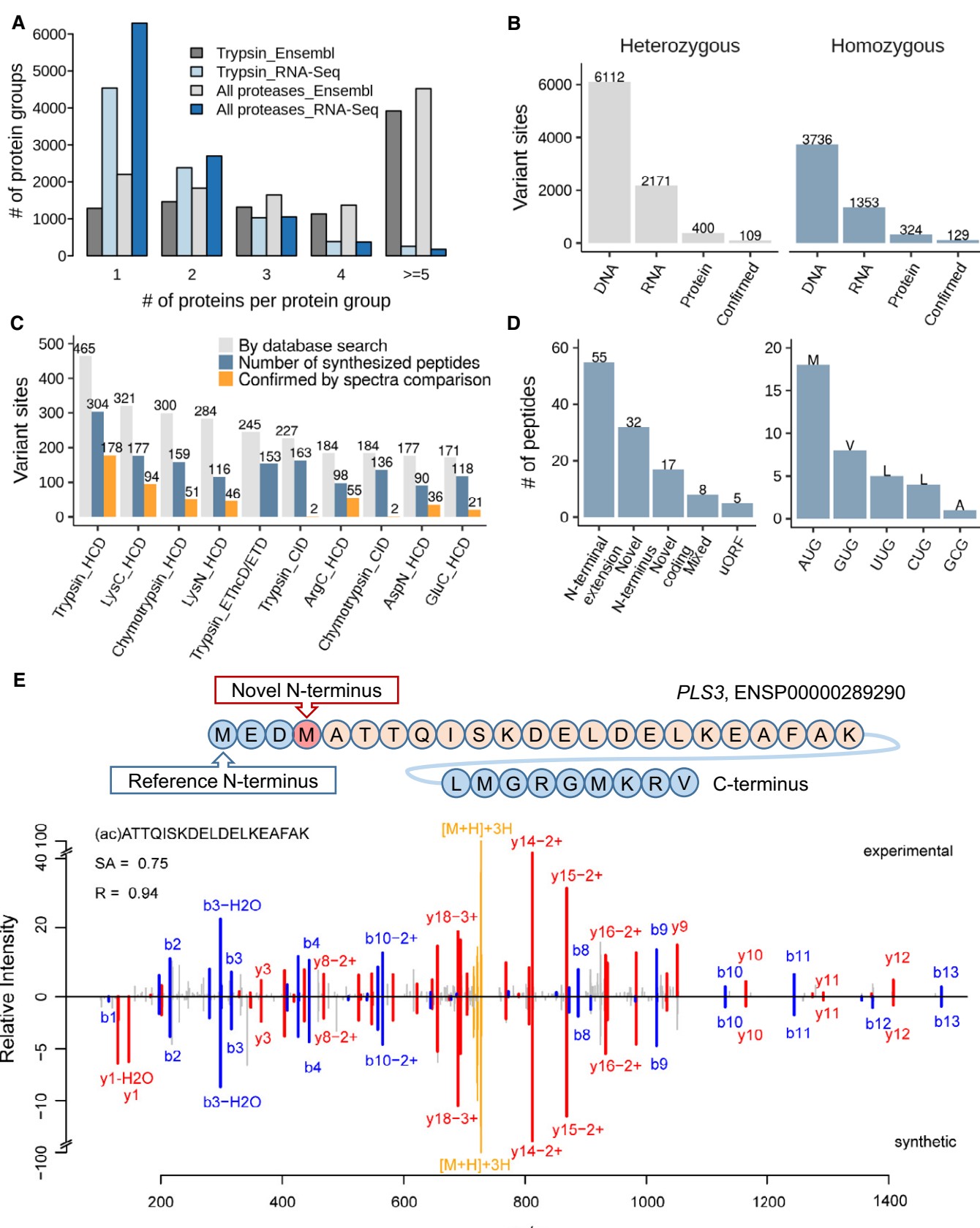

**Figure 4.**

◀

**Figure 4.  Proteogenomics exploration for protein-level detection of isoforms, single amino acid variants and alternative translation sites.**

A   Searching the tonsil proteomic data (trypsin alone or all enzymes) against a tissue-specific sequence database constructed from RNA-Seq data drastically reduces the number of individual protein sequences in protein groups compared to searches against Ensembl, allowing for the more efficient detection of protein isoforms.

B   Number of single amino acid variants detected by whole exome sequencing and DNA, by RNA-Seq at the mRNA and by mass spectrometry at the protein level as well as confirmed candidates by validation using synthetic peptide spectra comparisons. It is apparent that only a very small fraction of all variants detected at the DNA or RNA level can be detected at the proteome level using current technology.

C   Analysis of which proteomic workflow contributed to the detection and confirmation of single amino acid variants.

D   Results of the detection of non-canonical coding regions using proteomics data (left panel) and different alternative start codons identified by acetylated N-terminal peptides (right panel). The majority of cases are N-terminal extensions of annotated genes. All but one of the detected alternative translation start sites correspond to point mutations of the first base of the classical AUG codon.

E   Validation of a novel translation start site for the protein PLS3. The upper panel shows the novel translation site position within the amino acid sequence context, and the lower panel shows a mirror plot of the tandem mass spectra of the endogenous N-terminally acetylated peptide (peaks pointing upwards) and the corresponding synthetic peptide spectrum (peaks pointing downwards). Y-type sequence ions are coloured in red, b-type ions in blue, and the intact peptide as well as neutral losses thereof are marked in yellow.

underlying transcript in at least one of the tissues (FPKM > 1) and to fail a BLAST search against UniProt to exclude obvious alternative explanations. This approach yielded 5 lncRNA and 344 aTIS peptides, respectively. Because of the size of the searched database (aTIS: 474,991 entries; lncRNA: 29,524 entries), there was still ample opportunity to generate false positives. Interestingly, not a single lncRNA peptide could be substantiated by synthetic peptides indicating that lncRNA is rarely if at all translated (Bánfai *et al*, 2012).

To validate the candidate aTIS peptides, we compared spectra of endogenous and synthetic peptide reference standards as described above. Only 66 aTIS peptides (including 8 N-terminally acetylated peptides) covering 53 genes and 57 alternative translation start sites could be confirmed in this way (Table EV8). Manual spectrum interpretation yielded 96 aTIS peptides (overlap of 45 to the SA analysis) mapping to 76 genes and 81 alternative translation start sites. In total, we confirmed 117 aTIS peptides mapping to 89 genes and 99 alternative translation start sites, which included 14 peptides from 12 genes reported in previous studies, for example FXR2, RPA1 and CDV3 (Table EV8; Branca *et al*, 2014; Kim *et al*, 2014). Fifty-five of the above aTIS peptides represent 5′ N-terminal extensions of the original gene, 32 peptides represent novel (acetylated) N-termini downstream of the canonical start site, 17 represent frame-shifts potentially leading to an entirely new sequence, five peptides likely represent upstream ORFs (uORF) with a stop codon before the canonical start site and 8 peptides with mixed annotation (Fig 4D, left panel). The mirror mass spectra in Fig 4E for the endogenous (top) and synthetic (bottom) peptide (ac)ATTQISKDEL-DELKEAFAK from the actin-binding protein plastin-3 (PLS3) provide an example for the detection of a novel N-terminal sequence. For 36 of the peptides representing aTIS, we identified the exact start site as the peptide was N-terminally acetylated (Fig 4D, right panel). Among these, 18 contained an AUG start codon (Met), 8 contained a GUG start codon (Val), 5 a UUG and 4 a CUG start site (both Leu), and one GCG start site (Ala). This confirms the emerging notion that non-AUG translation initiation events are not as infrequent as previously thought and may represent a mechanism to regulate protein expression (Kearse & Wilusz, 2017). This study only identified a relatively small number of aTIS events compared to others (Na *et al*, 2018) implying that enrichment of N-terminal peptides (Gevaert *et al*, 2003; Kleifeld *et al*, 2010) is a more efficient way to detect such events systematically but also pointing out that the previous literature may not be free of a substantial number of mistakes.

An important learning from the present systematic analysis of transcriptomes and proteomes of human tissues is that identifying protein SAAVs or novel coding sequences using proteomics is possible but remains challenging. There were large discrepancies between the results of the two database search engines used (Mascot and Andromeda; Fig EV3F and G; Appendix Figs S20 and S21) implying that the underlying scoring schemes are not optimized yet for the detection of variants and novel coding regions. At present, synthetic peptide reference spectra appear to be mandatory for validation and manual spectra comparisons still have a role to play (Lee *et al*, 2018). Neither approach has been followed systematically in the literature so far and, obviously, they are also not without error but clearly more powerful than purely relying on statistical criteria with largely arbitrary cut-offs alone (Nesvizhskii, 2014; Dimitrakopoulos *et al*, 2016; Lee *et al*, 2018). It appears that even with the latest proteomic technology, proteogenomics currently offers rather small returns on very significant efforts in data generation, analysis and validation and that large improvements will be required to change this situation substantially in the future. It is possible that our filtering criteria were perhaps too strong so that further variants may be present in the data (see Table EV8). However, no convincing false discovery rate estimation has been published yet for spectral angle analysis (let alone for manual data analysis); hence, we decided to be conservative. Still, the resource presented in this work should be of considerable value for scientists wishing to develop more sophisticated approaches for proteogenomics in the future and the authors think that there is considerable future potential in the use of synthetic peptide references in conjunction with spectral angle analysis particularly for the many chimeric spectra present in classical data-dependent proteomic datasets but even more so for the increasing application of data-independent data acquisition regimes.

## Materials and Methods

### Human tissue specimen

The 29 human tissue samples used for mRNA and protein expression analysis were obtained from the Department of Pathology, Uppsala University Hospital, Uppsala, Sweden, as part of the sample collection governed by the Uppsala Biobank (www.uppsalabiobank.uu.se/en/). All tissue samples were collected and handled using standards developed in the Human Protein Atlas (www.proteinatlas.org)

   

and in accordance with Swedish laws and regulations. Tissue samples were anonymized in agreement with approval and advisory reports from the Uppsala Ethical Review Board (References # 2002-577, 2005-338 and 2007-159 (protein) and # 2011-473 (RNA)). The need for informed consent was waived by the ethics committee. The list of all tissues along with corresponding donor information, sample preparation and measurement information is provided in Table EV1.

## RNA sequencing

Procedures for RNA extraction from tissues, library preparation and sequencing have already been described (Uhlén *et al*, 2015). Briefly, pieces of frozen human tissue were embedded in optimal cutting temperature (OCT) compound and stored at −80°C. Cryosections were cut and stained with haematoxylin-eosin for microscopical confirmation of tissue quality and proper representativity. 5–10 cryosections (10 μm) were transferred to RNAse-free tubes for extraction of total RNA using the RNeasy Mini Kit (Qiagen). RNA quality was analysed with an Agilent 2100 Bioanalyzer system with the RNA 6000 Nano LabChip Kit (Agilent Biotechnologies). Only samples of high-quality RNA (RNA integrity number ≥ 7.5) were used for mRNA sample preparation and sequencing. The mRNA strands were fragmented using Fragmentation Buffer (Illumina), and the templates were used to construct cDNA libraries using a TruSeq RNA Sample Prep Kit (Illumina). Gene expression was assessed by deep sequencing of cDNA on Illumina HiSeq 2000/2500 system (Illumina) for paired-end reads with a read length of 2 × 100 bases. RNA sequencing data were aligned against the human reference genome (GRCh38, v83) using Tophat2.0.8b. FPKM (fragments per kilobase of exon model per million mapped reads) values were calculated using Cufflinks v2.1.1 as a proxy for transcript expression level. The FPKM values of each gene were summed up in an individual sample, and median normalization was applied to evaluate genes expression levels between tissues. A cut-off value of 1 FPKM was used as a lower limit for detection across all tissues.

## Sample preparation and off-line hydrophilic strong anion chromatography (hSAX)

Fresh frozen human tissue samples (parallel cryosections cut simultaneously as those used for RNA extraction, described above) were prepared for LC-MS/MS as described previously (Ruprecht *et al*, 2017). Briefly, tissue slices were homogenized in lysis buffer (50 mM Tris/HCl, pH 7.6, 8 M urea, 10 mM tris(2-carboxyethyl) phosphin hydrochloride, 40 mM chloroacetamide, protease and phosphatase inhibitors) by bead milling (Precellys 24, Bertin Instruments, France; 5,500 rpm, 2 × 20 s, 10 s pause). Protein content was determined using the Bradford method (Coomassie (Bradford) Protein Assay Kit, Thermo Scientific), and 300 μg of the protein extract was used for in-solution digestion with trypsin. For this, the sample was diluted with 50 mM Tris/HCl to a final urea concentration of 1.6 M, and trypsin was added at a 50:1 (w/w) protein-to-protease ratio. After 4 h of digestion at 37°C, another aliquot of trypsin was added to reach a final 25:1 (w/w) protein-to-protease ratio and the sample was incubated overnight at 37°C. In addition, the tonsil sample was subjected to digestion using LysC, ArgC,

GluC, AspN, LysN and chymotrypsin (LysC was from Wako, Japan; the other proteases were from Promega, USA). 300 μg of the protein extract prepared as described above was applied in each digestion. The buffers were prepared according to the manufacturer's protocols. The resulting peptides were desalted and concentrated on C18 StageTips (Rappsilber *et al*, 2007) and fractionated via hSAX off-line chromatography exactly as described previously (Ruprecht *et al*, 2017). The details of digestion for each tissue are given in the Appendix Table S1.

## On-line liquid chromatography–tandem mass spectrometry (LC-MS/MS)

Quantitative label-free LC-MS/MS analysis was performed using a Q Exactive Plus mass spectrometer (Thermo Fisher Scientific, Bremen, Germany) coupled on-line to a nanoflow LC system (NanoLC-Ultra 1D+, Eksigent, USA). Peptides were delivered to a trap column (0.1 × 2 cm, packed with 5 μm ReproSil-Pur AQ, Dr. Maisch GmbH, Germany) at a flow rate of 5 μl/min for 10 min in 100% solvent A (0.1% formic acid, FA, in HPLC-grade water). After 10 min of loading and washing, peptides were transferred to a 40 cm (75-μm inner diameter) analytical column, packed with 3 μm, ReproSil-Pur C18-AQ, Dr. Maisch GmbH, Germany) and separated using a 110-min gradient from 2% to 32% solvent B (0.1% FA, 5% dimethyl sulfoxide in acetonitrile, ACN) at a flow rate of 300 nL/min. Full scans (*m/z* 360–1,300) were acquired at a resolution of 70,000 using an AGC target value of 3e6 and a maximum ion injection time of 100 ms. Internal calibration was performed using the signal of a DMSO cluster as lock mass (Hahne *et al*, 2013). Tandem mass spectra were generated for up to 20 precursors by higher-energy collisional dissociation (HCD) using a normalized collision energy of 25%. The dynamic exclusion was set to 35 s. Fragment ions were detected at a resolution of 17,500 using an AGC target value of 1e5 and a maximum ion injection time of 50 ms.

LysC-, ArgC-, GluC-, AspN-, LysN- and chymotrypsin-digested samples were measured on a Q Exactive HF mass spectrometer (Thermo Fisher Scientific, Bremen, Germany) coupled on-line to a nanoflow LC system (NanoLC-Ultra 1D+, Eksigent, USA). Full scan MS spectra were acquired at 60,000 resolution and a maximum ion injection time of 25 ms. Tandem mass spectra were generated for up to 15 peptide precursors and fragments detected at a resolution of 15,000. The MS2 AGC target value was set to 2e5 with a maximum ion injection time of 100 ms. The other settings were the same as for the Q Exactive Plus.

Tryptic peptides from the tonsil sample were also analysed on an Orbitrap Fusion Lumos Mass Spectrometer (Thermo Fisher Scientific, Bremen, Germany) coupled on-line to a nanoflow LC system (UltiMate™ 3000 RSLC Nano System, Thermo Fisher Scientific) using CID, and EThcD/ETD fragmentation. Full MS scans were performed at a resolution of 60,000, a maximum injection time of 50 ms and an AGC target value is 5e5, followed by MS2 events with a duty cycle of 2 s for the most intense precursors and a dynamic exclusion set to 60 s. CID scans were acquired with 35% normalized collision energy and Orbitrap readout (1e5 AGC target, 0.25 activation Q, 20 ms maximum injection time, inject ions for all available parallelizable time enabled, 1.3 *m/z* isolation width). EThcD/ETD scans used charge-dependent parameters: 2+ precursor ions were

fragmented by EThcD with 28% normalized collision energy and 3+ to 7+ precursor ions were fragmented by ETD. The MS2 scans were read out in the Orbitrap (1e5 AGC target, 0.25 activation Q and 100 ms maximum injection time).

## MS data processing and database searching

For peptide and protein identification and label-free quantification, the MaxQuant suite of tools version 1.5.3.30 was used. The spectra were searched against the Ensembl human proteome database (release-83, GRCh38) with carbamidomethyl (C) specified as a fixed modification. Oxidation (M) and acetylation (protein N-Term) were considered as variable modifications. Trypsin/P was specified as the proteolytic enzyme with 2 maximum missed cleavages. The match between runs function was enabled, with match time window set to 0.7 min and an alignment time window of 20 min. The FDR was set to 1% at both PSM and protein level. LysC/P, ArgC and LysN were specified with two maximum missed cleavages. Searches for GluC and AspN peptides allowed three missed cleavages. Chymotrypsin (C terminal of F, Y, L, W or M) was allowed with at most 4 missed cleavages. Label-free quantification was performed using the iBAQ approach (Schwanhäusser *et al*, 2011). For non-tryptic peptides and single tissue analysis, matching data between fractions were disabled.

## Quantitative analysis of transcriptomes and proteomes

The quantitative analyses of proteomic and transcriptomic data were performed at the gene level. Since some genes have alternative Ensembl gene IDs with identical sequences, redundant protein sequences derived from these genes exist in protein sequence databases. These alternative gene IDs were not part of the reference sequences for the analysis of RNA-Seq data. To improve comparability between RNA and protein measurements, such identifiers were removed from the MaxQuant output files an updated protein ID and gene ID column was added to Table EV1. To evaluate gene expression level, the total abundance of each gene in all individual sample was used. The data were log-transformed (base 10) and normalized using median centring across tissues.

The genes were classified into "Tissue enriched", "Group enriched", "Tissue enhanced", "Expressed in all" and "Mixed" as described by Uhlén *et al* (2015, 2016). Gene ontology analysis of genes only identified in transcriptomes and proteomes, and the elevated proteins expressed in each tissue were performed using the R package "clusterProfiler" and *P*-values were adjusted according to the method by Benjamini–Hochberg (BH; Yu *et al*, 2012). The resulting (redundant) gene ontology terms (biology process) of elevated genes were removed using the "simplify" function in clusterProfiler based on GOSemSim (Yu *et al*, 2010). The list of 1,158 mitochondrial genes was obtained from MitoCarta 2.0 (Calvo *et al*, 2016). Essential genes ($n$ = 583) were assembled from three human essential gene studies using CRISPR-Cas9 and retroviral gene-trap genetic screens (Blomen *et al*, 2015; Hart *et al*, 2015; Wang *et al*, 2015). Diseases-related genes ($n$ = 3,896) and kinase genes ($n$ = 504) were obtained from UniProt. Cancer genes ($n$ = 719) were downloaded from Cosmic (Futreal *et al*, 2004). Drug target genes ($n$ = 784) were obtained from DrugBank (Wishart *et al*, 2018) and restricted to proteins directly related to the mechanism of action for at least one of the associated drugs. GPCR genes ($n$ = 1,410) were obtained from HGNC, and phosphatase genes ($n$ = 238) were from DEPOD (Duan *et al*, 2015). Transcription factor genes (IF, $n$ = 1,639) were obtained from the HumanTFs collection (Lambert *et al*, 2018).

The Spearman correlation coefficient was used for correlating transcriptome and proteome levels in single tissues. The slopes were estimated by ranged major-axis (RMA) regression, which allows errors in both variables and is symmetric, using the R package "lmodel2" (Csárdi *et al*, 2015). The protein–mRNA Spearman correlation coefficients of 9,870 genes which were at least expressed in 10 (20, 29) tissues at both mRNA and protein levels were calculated. The co-inertia analysis (CIA) was performed using the "cia" function in the "made4" R package (Culhane *et al*, 2005). A total of 9,870 genes which were expressed in at least 10 tissues at both mRNA and protein levels were considered, and the remaining missing values were replaced with a positive value $1 \times 10^4$ times smaller than the lowest expression value in each dataset.

Protein copy numbers were calculated from intensity values according to the "proteomic ruler" approach (Wiśniewski *et al*, 2014). Transcript copy numbers were calculated from FPKM values based on the estimated total cellular RNA amount using the total intensity of ribosomal proteins (Wiśniewski *et al*, 2014) and the assumption that the cellular mRNA mass represents about 2% of the total cellular RNA mass (Melnikov *et al*, 2012).

## Construction of sample-specific protein sequence databases from RNA-Seq data

RNA sequencing data were aligned to the human reference genome (GRCh38, v83) using Tophat2.0.8b. FPKM values were calculated using Cufflinks v2.1.1 as a proxy for transcript expression level. Rvboost was used for variant calling. All transcripts with FPKM > 1 were translated into protein sequences and included in the search database. Each tissue was searched against its matched RNA-Seq database using MaxQuant as described above. The match between runs function was disabled. The MaxQuant output data were used for the isoform analysis.

## Exome sequencing and variant calling

The exome of tonsil tissue was enriched using the Agilent SureSelectXT Kit (v5) and sequenced on an Illumina HiSeq 4000 sequencer. The raw data were aligned to the human reference genome (hg38) using bwa (v0.7.12), and duplicate reads were marked using Picard Tools (v2.4.1). Genomic variants were called and filtered using the GATK (v.3.6) HaplotypeCaller and VariantFiltration modules, respectively, according to the best practice guide (https://software.broadinstitute.org/gatk/best-practices/). Furthermore, variants at sites with a read depth < 10× were removed. We also removed any I/L variation as these cannot be distinguished by mass spectrometry. The resulting variants were annotated using the Ensembl Variant Effect Predictor (v85). The RNA sequencing data were aligned to the human reference genome (hg38) using STAR aligner (v2.5.2), and duplicate reads were marked using Picard Tools (v2.4.1). Variants were called using the GATK (v.3.6) HaplotypeCaller module, according to the aforementioned best practice guide.

A variant fasta formatted database was created by the "custom-ProDB" package from the exomic variants (Wang & Zhang, 2013). Mascot searching of the ultra-deep mass spectrometry data was performed against this database together with protein databases from UniProt and Ensembl using the following parameters: peptide mass tolerance set at 10 ppm, MS/MS tolerance set at 0.05 Da, carbamidomethylation of cysteine defined as fixed modification, oxidation of methionine and acetylation defined as variable modification. Trypsin-, LysC-, ArgC- and LysN-digested peptides allowed up to two missed cleavages. AspN- and GluC (V8-DE in Mascot search engine)-digested peptides with up to three cleavages were considered. Chymotrypsin-digested peptides were allowed to have a maximum of four missed cleavages. Resulting PSMs were analysed using Percolator (v3.01), and an overall FDR cut-off of 1% was applied.

A custom python script was used to identify PSMs covering variant sites and showing either the variant or the canonical genotype. All initial candidate variant peptides had met the following criteria: (i) Mascot ion scores of at least 25; (ii) a Mascot delta score of at least 10; (iii) the peptide must only map to the variant database; (iv) the peptide must map to a single genomic position only; (v) for missense variants, the peptide must either show the variant amino acid or it must be cleaved according to a novel protease cleavage site arising from the variant; and (vi) for nonsense variants, the peptide must end at the novel C-terminus. For canonical genotypes, the same criteria were applied except: (i) at least one protein the peptide maps to must not be from the variant database; (ii) for missense variants, the peptide must show the wild-type amino acid; and (iii) for nonsense variants, the end of the peptide must be after the novel C-terminus (after nonsense variant sites). The resulting candidate peptides were mapped against UniProt using BLAST to exclude other obvious explanations. To further consolidate the variants peptides and to reduce false positives, peptide identification by MaxQuant was performed in parallel. The customized exomic variant database was searched using the same parameters used for Ensembl database searches described above. The list of candidate variant peptides for the spectra angle analysis required the identification by both Mascot and MaxQuant.

### Identification of peptides translated from non-coding regions

A database of products from possible alternative translation initiation sites (aTIS) was constructed by searching the 5′ UTR of GENCODE transcripts (v25) for putative alternative start codons and *in silico* translating these "novel coding sequences". This resulted in 474,991 aTIS "proteins" > 6 amino acids. The lncRNA protein databases were generated by three-frame-translating the GENCODE (v25) lncRNA database, resulting in 29,524 sequences. The standard 29 tissue proteomics datasets were supplemented with two tissues for which only proteome data were available (bone marrow, pituitary gland); in total, 50 samples (including replicates of some organs) were searched against concatenated sequence collections comprising the aTIS and lncRNA databases, GENCODE (v25), UniProt (downloaded on 03 February 2017) and sample-specific RNA-Seq-based databases using Mascot to identify peptides from known proteins. The search parameters were the same as described for the exome variant peptide identification. The resulting PSMs were processed using Percolator, and an overall FDR cut-off of 1% was applied. A custom python script was used to identify PSMs from putative-translated lncRNAs or aTIS the database. Candidate peptides had to meet the following criteria: (i) the PSM must map to a single database only, i.e. aTIS or lncRNA but no any other; (ii) the Mascot score must be at least 25; (iii) the Mascot delta score must be at least 10; and (iv) the original underlying transcript must be expressed in at least one of the tissues (RNA-Seq FPKM > 1). The resulting PSMs were then mapped against UniProt using BLAST to exclude other explanations for the novel peptide (e.g. peptides arising from a novel tryptic cleavage site due to a genomic variant). To consolidate the list of candidate aTIS and lncRNA peptides and to reduce false positives, the raw MS data were also searched by MaxQuant (using the same parameters as described for searches using Ensembl). Only those peptides were allowed to pass to the stage of spectral contrast angle analysis if they were identified by both Mascot and MaxQuant.

### Validation of variant and non-coding peptides by synthetic reference peptides

All peptides which passed the filter criteria for Mascot described above were synthesized at JPT Berlin using Fmoc-based solid-phase synthesis. The details of peptide synthesis, sample preparation and MS measurement were as described (Zolg *et al*, 2017). Normalized spectral contrast angle (SA) analysis was performed to compare endogenous and synthetic peptides using in-house Python scripts (Toprak *et al*, 2014). Candidates passed if (i) they showed SA values of ≥ 0.7 (Pearson of ~0.9), (ii) the endogenous peptide had a Mascot score of 50 or higher or (iii) manual spectrum inspection substantiated the candidate peptide sequence assignment. In parallel, the tandem MS spectra of all candidate peptides were also inspected manually. For the identification of "missing proteins", we required an Andromeda score of ≥ 100. The other criteria were the same as above.

## Data availability

Transcriptome sequencing and quantification data are available in following database: RNA-Seq data: ArrayExpress E-MTAB-2836 (http://www.ebi.ac.uk/arrayexpress/experiments/E-MTAB-2836/). The raw mass spectrometric data and the MaxQuant result files have been deposited to the ProteomeXchange Consortium via the PRIDE partner repository (Vizcaíno *et al*, 2016; https://www.ebi.ac.uk/pride/archive/projects/PXD010154) with the dataset identifier: project accession: PXD010154.

**Expanded View** for this article is available online.

### Acknowledgements

This work was in part funded by the German Excellence Initiative cluster Center for Integrated Protein Analysis Munich (CIPSM). This work was in part supported by the Knut and Alice Wallenberg Foundation. The authors wish to thank pathologists and staff at the Department of Clinical Pathology, Uppsala University Hospital, for providing the tissues used in the study, Harald Marx for guidance on the construction of protein sequence databases and Professor Marily Theodoropoulou for providing pituitary

samples. DW is grateful for a scholarship from the China Scholarship Council. BE and JG are supported by EU Horizon2020 Collaborative Research Project SOUND. A fellowship by the Graduate School of Quantitative Biosciences Munich (QBM) supports BE.

## Author contributions

BK, FP, HH, JG and MU conceived and designed the study. FP selected and provided normal human tissue samples. AA, DW, DPZ, KS, JZ and LL performed experiments. BE, BH, BK, CM, DW, HH, JG, MF, MW, TH, TS and TW performed data analysis. BE, BK, DW, HH, JG and TW wrote the manuscript.

## Conflict of interest

HH and TH are employees of OmicScouts GmbH. HH, MW and BK are co-founders and shareholders of OmicScouts GmbH. MW and BK have no operational role in OmicScouts GmbH. KS is employee of JPT Peptide Technologies.

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
