## [Review Process File · Molecular Systems Biology]

A deep proteome and transcriptome abundance atlas of 29 healthy human tissues

Dongxue Wang, Basak Eraslan, Thomas Wieland, Björn Hallström, Thomas Hopf, Daniel Paul Zolg, Jana Zecha, Anna Asplund, Li-hua Li, Chen Meng, Martin Frejno, Tobias Schmidt, Karsten Schnatbaum, Mathias Wilhelm, Frederik Ponten, Mathias Uhlen, Julien Gagneur, Hannes Hahne and Bernhard Kuster.

Review timeline:

Submission date:	20 th June 2018
Editorial Decision:	10 th August 2018
Revision received:	16 th November 2018
Editorial Decision:	18 th December 2018
Revision received:	1 st January 2019
Accepted:	8 th January 2019

Editor: Maria Polychronidou

Transaction Report:

1st Editorial Decision

10th August 2018

Thank you again for submitting your work to Molecular Systems Biology. We have now heard back from the four referees who agreed to evaluate your study. Reviewers #1, #2 and #3 have also evaluated the related manuscript MSB-18-8503. As you will see below, the reviewers think that the study presents a valuable resource for the community. They raise however a series of concerns, which we would ask you to address in a major revision.

Overall, I think that the reviewers' recommendations are clear and there is therefore no need to repeat the points listed below. Reviewer #2 provides several constructive suggestions on some additional analyses that would increase the impact of the study. Please feel free to contact me in case you would like to discuss in further detail any of the issues raised by the reviewers.

REFERE REPORTS

Reviewer #1:

Summary

In this manuscript, the authors measured matched quantitative proteomics and transcriptomics of 29 healthy human tissues to a depth of about 11,000 protein quantifications and 12,000 transcript quantifications per tissue on average. The authors briefly examine tissue specificity, as well as expressed transcripts with no protein identification, quantified proteins whose mRNA transcript are not detected, and more generally the transcript to protein relationship. Finally, an extensive proteogenomics characterization in terms of RNAseq based isoform-specific database, proteomics evidence for single amino acid variants, alternative translation initiation sites and proteins from long non-coding RNAs, is presented.

General remarks

The manuscript presents a carefully generated large data set, which would be a valuable resource for analysis investigating the transcript-proteins relationship. Beyond an obligatory basic characterization the proteogenomic analyses are very soundly conducted, findings validated through synthetic peptides and results not overstated. The manuscript does however, have limited direct findings and conclusions and a few points (see below) should be addressed before publication.

Contrasting the two manuscripts:

The Wang et al (MSB-18-8503) manuscript in contrast to the Eraslan et al (MSB-18-8513) manuscript is well written, has high quality figures & supporting figures, etc. The Wang/8503 manuscript is publishing the generated data incl. the methods description and as such presents a classical data resource paper with a sound proteogenomics analysis but limited findings and conclusions. In contrast, the Eraslan/8513 manuscript is not only not a resource manuscript, but also presents a way less sound analysis, to the point where it's reliability is unclear, and even if true limited finding and impact.

Fig. 1A,B,C in MSB-18-8513 are basically equal to Fig. 1A, 2A, and 2B of MSB-18-8503, with the ones in MSB-18-8503 being generally better. Also note that for the same data 1C and MSB-18-8503 2B reported different regression slopes 2.76 and 2.6 respectively.

Specific points:

1. Page 6, "it also provides protein level evidence for 72 proteins (represented by at least one unique peptide with Andromeda score of > 100) that are not yet covered by neXtProt". I cannot reproduce this from Table EV1 (tab "F"): I am counting 26 "Leading razor proteins" with these criteria (Score > 100, Unique (Protein) = yes). I also don't understand why this list includes trypsin (CON__P00761), which is of course not a missing protein.
2. Page 8, "The much wider dynamic range at the protein level implies that protein synthesis and protein stability play an important role [...] beyond mRNA levels. Moreover, the number of protein copies produced per molecule mRNA appear to be much larger for high- than for low-abundance transcripts [...]". This was observation was made and discussed in some detail before and is thus not new.
3. Fig. 3 B: The figure contains light grey lines. It is not clear what they are supposed to indicate.
4. Figure 2A is somewhat misleading since mRNA and proteins are per definition not in same units (since they are FPKM and iBAQ, but would only be if molecules or something similar could be calculated), but here put on one axis.
5. Page 9 "Due to the fact that a majority of the proteins are expressed at similar levels across human tissue, it is not very surprising that the correlation of mRNA/protein ratios across tissues is generally not very strong (Fig 2E; median 0.35)". Small variability does not prevent high or significant correlations and this is independent of the scaling. However, indeed the correlation coefficient presented 0.35 seems quite low in comparison to published cancer studies with match transcriptomics and proteomics. Is this due to using the iBAQ values (made for within sample comparisons) for the across tissues comparison - How is the correlation when using LFQ values?
6. Page 10, "However, when searching the same data against a protein sequence database constructed from the tissue specific RNA-Seq data [...] the proportion of single entry protein groups increased to 53%". The interpretation of this observation is not entirely clear: How good is the annotation of different isoforms in the tissue-specific RNAseq data? If this annotation systematically misses isoforms it will of course reduce the number of proteins per protein group.
7. Page 11 "no clear consensus in the proteomics and transcriptomics communities as to how

quantitative values should be allocated to particular proteins or transcripts [...]It would not be surprising if these differences in quantification approaches would add substantially to the poor correlation of mRNA and protein levels (or their ratios)." - Given this observation, what are the reported correlations if the parsimonious approach is applied to both proteomics and transcriptomics? And if the distribution approach is applied to both?

8. Page 12: Why are suddenly also UniProt sequences included in the search in addition to Ensembl? This should be consistent with the other analysis or the reason for the deviation should be given.

9. Page 13: "Instead, the main reasons for poor coverage of variants at the proteome level are [...]" - The possibility that these variants are truly not existing at the protein level (e.g. because of sequencing/calling errors, they not being translated, they being rapidly degraded thus not measurable at steady state,...) should be discussed here as well.

Reviewer #2:

Wang et al. present a manuscript on quantitative RNA and protein expression measurements across 29 healthy human tissues. The authors discuss major trends across the >11,000 genes in the main dataset. The manuscript represents a valuable resource for the community as its depth and controlled character are unprecedented. The authors illustrate this value by highlighting several trends the observe in the work. A more detailed discussion of sequence and other characteristics follows in another manuscript.

Overall, I am excited about the work, but would like to see several questions and criticisms addressed prior to supporting its acceptance in the journal.

These criticisms primarily address the focus/emphasis of the current presentation. I feel the authors have all the data at their hands and digging a bit deeper into some new questions (rather than confirming known trends) will move the work from being a resource to also being an insightful paper.

MAJOR

Quality control.

With the in-depth quantitation of the tonsil tissue, the authors missed an opportunity to provide an assessment of the overall quality of more screening-based approaches. What I mean: how much do the protein concentrations measured in the simple shotgun vs. the multi-protease approach correlate? What does that tell us about label-free quantitation in the proteomics experiments conducted for 29 tissues? How reliable are they? Can we do that or not or should we go for in-depth?

Same applies to RNA level - with the tonsils, the authors have the unique chance to assess if the normal RNA-seq approach provides accurate estimates of abundance - or if not, what is the range in reliability?

The authors published themselves in this very journal work on protein-RNA correlations across cell lines (Edfors, 2016) which should also be cited. The work here extends this work. However, since the 2016 paper was done on highly accurate, targeted RNA and protein measurements, again, it can perhaps be used to QC/VALIDATE the more shotgun/larger-scale quantitation that is done here. Same applies to comparison to Wilhelms 2014 data.

Further, is there ANY data on an assessment of reproducibility of the findings? I did not find it, not for the tonsil, nor the other tissues. At least some assessment of reproducibility of quantitation or identification is crucial for anyone who wants to use this enormous dataset but doesn't know to which extent the quantification is reliable.

Quantitation of the expressed proteome, tissue-specific and shared proteins.

As the authors state, the extent of brain/testis specific proteins isn't new - but the authors again have the unique chance to validate some of these findings and the QUANTIFY how much of tissue-specific expression is due to our technological inability to detect the proteins (MS not sensitive enough), computational, or biological (i.e. true tissue-specific expression). I.e. for the in-depth tonsil data, how many of the originally tonsil-specific proteins were then detected in the in-depth protein

data? That should give an estimate of how many proteins are missed at the basic technology level. Further, there should be antibodies for some of the new or tissue specific proteins - how many are detected by western blots? Or targeted MS? - The authors don't need to test everything, but it's a shame to miss on the chance to actually quantify the extent of technological limitations and true biological trends. -

Again, same applies for RNA-level. For the few proteins that did not have detectable RNA levels, why did the authors not do some qPCR to see if the RNAs are truly not there? And if they are there, this would again give us a feel for technical limitations and perhaps, for the first time, a way to quantify these limitations.

What I mean - with this large dataset, the authors shouldn't miss out on the chance to provide a baseline dataset on the true tissue-specificity of RNA and protein expression, the true extent of missing RNA or proteins etc.

Detection of alternative splice variants at the protein level.

The authors go into some discussion of protein-level detection of the splice variants. The dominance of one isoform is not new. The authors then say they detect 41% of the variants by the in-depth/targeted proteomics approaches. I would say that this is actually a high fraction (and higher than previous attempts to MS-detect AS variants). I would like to see a more careful connection to existing work, again - rather than just confirming "Oh yes, it's low, we show that too" - I think the authors can move far beyond this and say "Using our in-depth proteomics approach we can indeed identify a large fraction of the splice variants".

Somewhat more curiosity but also along the lines of helping researchers know how good non-in-depth proteomics data is: how many splice variants are detected in the 'normal' tonsil data, that hasn't been analyzed in depth?

Correlation between and across RNA and protein concentrations.

In my opinion, this is the weakest part of the manuscript and can be shortened. The correlation has been extensively discussed elsewhere (including the authors' own papers) and does not provide new insight. On the contrary, I would argue that some of the presented results are somewhat out of date given other findings.

I encourage the authors strongly to incorporate Fortelny's findings (Nature 2017) into their discussions, along with Franks (PlosCB 2017) which is already briefly mentioned. Both papers show that the wider dynamic range of the protein concentrations explains much of what is often discussed as stronger correlations of protein concentrations across tissues compared to RNA concentrations across tissues. There is still a lot of confusion about this inside and outside the field, and with this potentially high-impact publication, the authors shouldn't miss a chance to clarify some of the misconceptions.

The Fortelny and Franks papers definitely need to be cited in the interpretation of the correlations line 252. It doesn't invalidate the authors findings, it just requires a shift in thinking that is important to convey to the community. While 'buffering' (discussed in next sentence) might seem more attractive with respect to the biological interpretation, it is important to clarify that much of the observed trends can be explained purely by differences in dynamic range. Ironically, the authors' own figure (figure 2F) nicely illustrates this effect (even though on a totally different sample). Given similar variation and similar numbers of datapoints, the 'squished' distribution on the right automatically has a lower correlation coefficient, DESPITE overall similar variation as the data on the left.

The statement in figure legend for Figure 3A is therefore entirely misleading. It suggests that this finding is new (which it isn't).

(While I find the correlation discussion overall too long, the example in Figure 2F is actually very nice and could be highlighted more. This would happen automatically if the other stuff is shortened.)

uTIS and non-canonical start codons.

This part is very interesting and can be highlighted more. By shortening other parts (see above) this will probably automatically happen.

MINOR

1. It might be a matter of taste, but many paragraphs are a page long or longer. They can easily be split and guide the reader.

2. Lines 208 onwards, I do not understand the logic in the second sentence: the dynamic range of transcripts detected by RNA-Seq spanned about four orders of magnitude and that of proteins detected by mass spectrometry spanned eight orders of magnitude (Fig 2A). This difference alone explains (at least in part) the overall higher coverage of the expressed proteome by RNA-Seq compared to that of LC-MS/MS.
3. Lines 291 - I find the fact that more protein groups with just 1 member are found in a database restricted to the RNA-seq data (instead of something that includes all variants) redundant and not surprising. The RNA-seq data will likely contain only one or few variants, so not surprising that fewer variants are then found using this as the basis for the proteomics search. This interpretation (unless I am missing something) can be entirely deleted.
4. Line 572 and corresponding figures: if the number of datapoint is low (like here, potentially just 10), correlation coefficients are sensitive to the number of datapoints. Therefore, the authors should probably not mix the discussion of data with 10 vs. all 29 datapoints. At the same overall variability, these different datasets will give different correlation coefficients (think of it this way: 2 points are perfectly correlated, always, you can always fit a straight line through 2 datapoints; three a bit less, four a bit less...).
5. Figure 1B - any way the data has been sorted?
6. Figure 1C - would be nice to discuss the high-abundance proteins not detected at RNA level, and again to check with qPCR!!!
7. Figure 3A - labeling of axes is confusing. And also, either this figure needs to be left out or discussed properly with the proper explanation and citations - the observed correlations are likely ENTIRELY due to the larger dynamic range of the proteomics data.
8. Figure 4C's message not entirely clear.

Reviewer #3:

The manuscript "A deep proteome and transcriptome abundance atlas of 29 healthy human tissues" by Wang et al reports the generation of RNA sequencing and proteomics data covering the expression of 17,615 transcripts and 13,664 proteins in 29 human tissues. To my knowledge this is the largest, most comprehensive single-study report of this kind. The matched tissue expression data are acquired from adjacent cryosections of the same samples, which allows for a direct comparison of mRNA and protein expression values. Therefore, I have no doubt that the data will be a useful resource to address a range of biological questions regarding gene expression. In addition, the manuscript reports a smaller proteomics dataset optimised for sequence coverage but demonstrates that even such state-of-the-art data are insufficient to power robust proteogenomics analyses on their own, highlighting the need for major technical and computational advances to make proteogenomics a reality.

I have no major concerns with regards to this paper, but several points should be clarified / addressed before considering it:

λ Sample description: It is unclear how many donors, replicates and actual samples were used. I think this should be described in somewhat more detail.

λ Line 121 and others: Is the number in brackets the standard deviation?

λ The missing testis proteome: This is a very interesting observation and I think the authors could elaborate a bit more on these unexpected findings. Are the 300 missing proteins from the HPA project randomly distributed across the RNA expression range or towards the lower end of the range? You mention the latter to be enriched in membrane proteins, possibly explaining a lack of extraction / detection at the protein level. What about the proteins from the other end of the spectrum, those where lack of detection is less likely to be related to their abundance. Are they enriched in any common function, or perhaps secreted? Are they very small?

λ Tissue enriched expression: The 4.3% tissue-enriched mRNAs contain still a substantial fraction of mRNAs only expressed in sperm but, as you suggest, many of those may not form stable proteins. So, one would expect the number of tissue-specific proteins to be even lower than that, but instead it is higher (5.4%). Could that reflect a sensitivity issue? In the sense that lack of detection of low abundance proteins in some tissues creates the appearance of tissue-specific expression and hence even these low numbers may still be an overestimation of true tissue-specific expression? This should be clarified or discussed.

λ Fig 2A: I find this plot misleading because it seems to show FPKM and iBAQs as if they were the same unit (the legend is not so clear). One would intuitively interpret it as showing that proteins are

more abundant than mRNAs. While that may be true, one can of course not infer that from comparing unrelated abundance measures.

λ Abundance differences: I generally agree with the author's conclusion that there appears to be huge discrepancy between the most abundant mRNAs and proteins in a tissue (Fig 2D). However, the examples shown in Fig 2C raise a few questions. First, it seems that the discrepancy is one-sided in the sense that the most abundant proteins (myosins) also rank very high at the mRNA level. I also noticed that the most abundant mRNA, MT-ATP8, encodes a tiny protein (7 kDa) - could the gene / mRNA / protein size confound the analysis, perhaps through the way FPKMs / iBAQs are calculated? Also, "MT" genes are not just mitochondrial proteins, but they are also encoded by the mitochondrial genome. Could the different processing and location of these mitochondrial mRNAs confound the RNA-seq analysis, i.e. overestimate their abundance relative to nuclear RNAs?

λ Fig 2E: It's unclear what this is showing. The text says correlation of mRNA / protein ratios across tissues. I assume that means each correlation coefficient reflects how well mRNA and protein abundances correlate across the 29 tissues, with a median of 0.35. And was that calculated on log₁₀ ratios? The figure legend does not mention mRNAs.

λ Fig 2F: This is a very nice example, but maybe the authors should replace the term "similar expression levels" (line 246/247) with a quantitative term, because people have different interpretations of what "similar" means.

λ Fig. 3B: I'm not sure about the co-inertia analysis (CIA). It is not a commonly used method. If the goal is to show that mRNA and protein abundances are more similar within than across tissues, a simple correlation analysis would probably be sufficient, e.g. colour in the diagonal in Fig. 3A and add the median values. The highlighted tissue groups are a quite selective and don't stand out visually as intuitive clusters. Again, for this purpose maybe PCA or tSNE may be more standard methods better suited for the task. Does CIA add anything else to this analysis? If there is nothing substantial I think the authors should use a more frequently used method to increase comprehensibility.

λ I could not log in to PRIDE with the details provided.

λ In addition to the raw files themselves, the processed data of the paper will be a great resource, but only if they are documented better and made available in more accessible form. For example, table EV1 appears to contain the key datasets. But what exactly does "genes in proteome" mean? Or "transcripts in transcriptome"? Are these FPKMs? Etc... It would also be good to provide the tables as txt or csv files rather than Excel.

λ Line 158: missing space

Reviewer #4:

This manuscript entitled "A deep proteome and transcriptome abundance atlas of 29 healthy human tissues" by Wang et al. provides an expression catalog of proteome and transcriptome from 29 paired healthy human tissues and discusses the expression differences within and across tissues. The authors describe the tissue enriched gene and protein expression and their role in studying the drug targets. Limitations of proteogenomics analyses to identify and validate the tissue specific isoforms and coding variants are also well described. Overall, I feel that this manuscript is well-written and is suitable for publication after the following minor concerns are addressed:

1. A description about the correlation of transcriptome from tissues that were also profiled in GTEx project should be provided. This will help assess if the presented differences across tissues were real or partial artifacts of tissue heterogeneity that can be confounded by factors such as sex and age.
2. This study demonstrates that about 50% of genes have elevated expression in one or more tissues. Because tissue gene expression can also be regulated by isoform switching, it would be informative if the authors could comment on the tissue-specific expression profile at the isoform level.
3. The authors should mention how many proteins out of 72 "missing" in neXtProt were identified that meet the HPP guidelines (Deutsch et al. 2016) to qualify for protein level evidence. They could also comment on their expression levels at transcriptome and proteome in tissues and if they are enriched in any tissues.
4. Are there common genes across tissues that are seen only at transcriptome level but not at the protein level? If yes, what class of processes/functions are they involved in?
5. The authors should show separate plots similar to Figure 3A for all five classes of tissue-specific

expression profiles to observe their patterns of transcriptome and proteome expression across the tissues.

1st Revision - authors' response

16th November 2018

A deep proteome and transcriptome abundance atlas of 29 healthy human tissues

Detailed response to reviewer comments:

The authors are grateful to the comments made by the reviewers. The new data, data analysis, figures and text have made the manuscript much stronger and the authors hope that all concerns have been adequately addressed.

Reviewer #1:

Summary

In this manuscript, the authors measured matched quantitative proteomics and transcriptomics of 29 healthy human tissues to a depth of about 11,000 protein quantifications and 12,000 transcript quantifications per tissue on average. The authors briefly examine tissue specificity, as well as expressed transcripts with no protein identification, quantified proteins whose mRNA transcript are not detected, and more generally the transcript to protein relationship. Finally, an extensive proteogenomics characterization in terms of RNAseq based isoform-specific database, proteomics evidence for single amino acid variants, alternative translation initiation sites and proteins from long non-coding RNAs, is presented.

General remarks

The manuscript presents a carefully generated large data set, which would be a valuable resource for analysis investigating the transcript-proteins relationship. Beyond an obligatory basic characterization the proteogenomic analyses are very soundly conducted, findings validated through synthetic peptides and results not overstated. The manuscript does however, have limited direct findings and conclusions and a few points (see below) should be addressed before publication.

Contrasting the two manuscripts:

The Wang et al (MSB-18-8503) manuscript in contrast to the Eraslan et al (MSB-18-8513) manuscript is well written, has high quality figures & supporting figures, etc. The Wang/8503 manuscript is publishing the generated data incl. the methods description and as such presents a classical data resource paper with a sound proteogenomics analysis but limited findings and conclusions. In contrast, the Eraslan/8513 manuscript is not only not a resource manuscript, but also presents a way less sound analysis, to the point where it's reliability is unclear, and even if true limited finding and impact.

The authors are happy to read that this reviewer thinks that the overall work is sound. We comment separately on the other study as part of the point to point response to that manuscript.

Fig. 1A,B,C in MSB-18-8513 are basically equal to Fig. 1A, 2A, and 2B of MSB-18-8503, with the ones in MSB-18-8503 being generally better. Also note that for the same data 1C and MSB-18-8503 2B reported different regression slopes 2.76 and 2.6 respectively.

These figures have been removed from the Eraslan et al manuscript. The minor differences in the regression lines in Figure 2B stem from differences in the normalization of the RNA-Seq data. As this does not change the conclusions of the analysis, we decided not to repeat the complete analysis from scratch and hope the reviewer finds this acceptable.

Specific points:

1. Page 6, "it also provides protein level evidence for 72 proteins (represented by at least one unique peptide with Andromeda score of > 100) that are not yet covered by neXtProt". I cannot

reproduce this from Table EV1 (tab "F"): I am counting 26 "Leading razor proteins" with these criteria (Score > 100, Unique (Protein) = yes). I also don't understand why this list includes trypsin (CON_P00761), which is of course not a missing protein.

We apologize for the oversight on our part. We have removed the “contaminant” proteins. To clarify, we performed the analysis on the gene level rather than on the isoform level as this reviewer appears to have done. We also checked the most recent release of nextprot and 67 proteins identified in the current study (Andromeda score of ≥ 100) had no protein evidence in nextprot (added to EV Table 1). Validation by synthetic peptides and requiring a spectral contrast angle of ≥ 0.7 reduced this number further to 37 proteins. We have updated the manuscript accordingly and are providing mirror plots for all synthetic peptide comparisons as part of the PRIDE submission.

2. Page 8, "The much wider dynamic range at the protein level implies that protein synthesis and protein stability play an important role [...] beyond mRNA levels. Moreover, the number of protein copies produced per molecule mRNA appear to be much larger for high- than for low-abundance transcripts [...]" This was observation was made and discussed in some detail before and is thus not new.

We acknowledge that this observation is not new *per se*. However, this study shows that this is a general phenomenon observed in all human tissues. We have revised the manuscript to make this clearer.

3. Fig. 3 B: The figure contains light grey lines. It is not clear what they are supposed to indicate.

Because the CIA plot is quite busy, the grey line just help connecting the tissue names to the corresponding arrows. We have clarified this in the figure legend.

4. Figure 2A is somewhat misleading since mRNA and proteins are per definition not in same units (since they are FPKM and iBAQ, but would only be if molecules or something similar could be calculated), but here put on one axis.

We have modified Figure 2A to indicate that the measure of abundance is iBAQ and FPKM. We also added a similar figure to the appendix that shows the distribution on the basis of copy numbers. Protein copies based on the proteomic ruler approach correlated very well to iBAQ values (Spearman $r=0.95$). The respective correlation for mRNA copies vs FPKM was not quite as good (Spearman $r=0.67$). Determining mRNA copies can only be done accurately by spiking standards. However, such data is not available for any mRNAs in our system. Therefore, we used an alternative approach published previously (PMID: 25225357, 22664983) that estimated mRNA abundance based on the observation that the mass of total mRNA is ~ 1 -3% of the mass of ribosomal proteins. This assumption is likely too simplified to yield accurate results for all mRNAs. Still, the conclusions we draw in the manuscript are not affected as there are still orders of magnitude in abundance differences between mRNA and protein levels the dynamic range of expression of RNA and protein within a tissue.

5. Page 9 "Due to the fact that a majority of the proteins are expressed at similar levels across human tissue, it is not very surprising that the correlation of mRNA/protein ratios across tissues is generally not very strong (Fig 2E; median 0.35)". Small variability does not prevent high or significant correlations and this is independent of the scaling. However, indeed the correlation coefficient presented 0.35 seems quite low in comparison to published cancer studies with match transcriptomics and proteomics. Is this due to using the iBAQ values (made for within sample comparisons) for the across tissues comparison - How is the correlation when using LFQ values?

Because iBAQ values correlate so well with copy numbers, we do not think that using LFQ would make a strong difference. Unfortunately, this cannot be experimentally tested because MaxQuant cannot handle $>1,000$ raw files in an LFQ-type of analysis even when run on a very powerful server computer (we tried). Further, the developers of MaxQuant/LFQ state themselves that LFQ should only be used for very similar samples. Given that the overall protein composition of the different tissues is not that similar, the LFQ approach would likely perform worse than iBAQ. As for the overall low correlation of 0.35, we clarify that this is not the correlation of mRNA and protein abundance in one single tissue but for one gene across many tissues. For the former, our data has

higher correlation values (between 0.42 and 0.58, see appendix). For the latter, we checked three datasets published by the CPTAC consortium and found median values of 0.23 for colon and rectal cancer tissues (PMID: 25043054), 0.39 for breast cancer tissues (PMID: 27251275) and 0.45 for ovarian cancer tissues (PMID: 27372738). Hence, it appears that our value is broadly in line with the literature. We note though that the CPTAC studies each focused on a single cancer type where one might expect higher correlation but our study included many different tissues which might further explain why the correlations are not higher than they are.

6. Page 10, "However, when searching the same data against a protein sequence database constructed from the tissue specific RNA-Seq data [...] the proportion of single entry protein groups increased to 53%". The interpretation of this observation is not entirely clear: How good is the annotation of different isoforms in the tissue-specific RNAseq data? If this annotation systematically misses isoforms it will of course reduce the number of proteins per protein group.

We agree that this can potentially happen. We used TopHat for isoform calling from the RNA-Seq data which is a mature tool but the software may miss some of the isoforms. We observed rare cases where we identified a protein (not necessarily an isoform) when searching Ensembl rather than the RNA-Seq data (e.g. because of the lower FPKM cutoff of ≥ 1 we required). But the effect observed in the reduction of proteins in a protein group is simply too large for this to be a reasonable explanation.

7. Page 11 "no clear consensus in the proteomics and transcriptomics communities as to how quantitative values should be allocated to particular proteins or transcripts [...] It would not be surprising if these differences in quantification approaches would add substantially to the poor correlation of mRNA and protein levels (or their ratios)." - Given this observation, what are the reported correlations if the parsimonious approach is applied to both proteomics and transcriptomics? And if the distribution approach is applied to both?

This is indeed a valid concern that we share and which requires further work in the future. Unfortunately, we were unable to address the question directly. This is because the software we used for RNA-Seq data processing allocates shared reads to sequences based on a probability measure derived from read length distributions. Other software tools have related ways of dealing with shared reads. We have, unfortunately, no way of calculating the 'parsimonious' data as requested because (as far as we know) it would require us to write new RNA-Seq processing software which is beyond the scope of the manuscript. As mentioned above, because iBAQ correlates well with data from spiked in standards and FPKM values correlate well with qPCR results, the issue may not be as severe as one might fear. But again, this is an interesting point that should be clarified at some point.

8. Page 12: Why are suddenly also UniProt sequences included in the search in addition to Ensembl? This should be consistent with the other analysis or the reason for the deviation should be given.

We used Uniprot simply to make sure that there were no better alternative explanations for the variant peptides we identified (as stated in the methods section).

9. Page 13: "Instead, the main reasons for poor coverage of variants at the proteome level are [...]" - The possibility that these variants are truly not existing at the protein level (e.g. because of sequencing/calling errors, they not being translated, they being rapidly degraded thus not measurable at steady state,...) should be discussed here as well.

We have expanded the discussion as requested.

Reviewer #2:

Wang et al. present a manuscript on quantitative RNA and protein expression measurements across 29 healthy human tissues. The authors discuss major trends across the >11,000 genes in the main dataset. The manuscript represents a valuable resource for the community as its depth and controlled character are unprecedented. The authors illustrate this value by highlighting several trends they observe in the work. A more detailed discussion of sequence and other characteristics follows in another manuscript.

Overall, I am excited about the work, but would like to see several questions and criticisms addressed prior to supporting its acceptance in the journal.

These criticisms primarily address the focus/emphasis of the current presentation. I feel the authors have all the data at their hands and digging a bit deeper into some new questions (rather than confirming known trends) will move the work from being a resource to also being an insightful paper.

The authors are happy to learn that this reviewer shares our excitement about the work.

MAJOR

Quality control.

With the in-depth quantitation of the tonsil tissue, the authors missed an opportunity to provide an assessment of the overall quality of more screening-based approaches. What I mean: how much do the protein concentrations measured in the simple shotgun vs. the multi-protease approach correlate? What does that tell us about label-free quantitation in the proteomics experiments conducted for 29 tissues? How reliable are they? Can we do that or not or should we go for in-depth?

We added information regarding all points raised on quality to the appendix rather than the main manuscript because we felt that inclusion in the main manuscript would disrupt the flow of the manuscript too much. More specifically, we correlated the total intensities of peptides for a given protein of the standard trypsin-HCD workflow either with the separate enzymes as well as the combination of all enzymes/MS/MS types. In summary, the correlations range from 0.71 (AspN-HCD vs Chymotrypsin-CID) to 0.93 for Trypsin-HCD vs Trypsin-ETcD/ETD. The highest correlation was obtained when comparing Trypsin-HCD vs all data ($r=0.94$) which is because the Trypsin-HCD workflow provided the richest data. From this we conclude that the standard Trypsin-HCD is still a reliable way to quantify proteins in complex mixtures. This is also reflected by the fact that the in-depth analysis did not identify vastly more protein coding genes than the Trypsin-HCD only workflow. These extra proteins covered the entire mRNA abundance range. In that sense, going in-depth does not seem to be required for the purpose of general quantitative proteomic profiling but has advantages when asking questions about isoforms etc. as we do in the manuscript.

Same applies to RNA level - with the tonsils, the authors have the unique chance to assess if the normal RNA-seq approach provides accurate estimates of abundance - or if not, what is the range in reliability?

We were unable to address the question directly as we could not independently determine copy numbers for mRNA (see above for how we estimated RNA copies). However, in an attempt to test reliability, we correlated FPKM values for the tonsil data with i) the total intensity of all proteases and ii) the total intensity for Trypsin-HCD only and obtained Spearman correlation coefficients of 0.56 and 0.54 respectively. These values are very close to each other and nearly identical to the correlation based on iBAQ values shown in a figure added to the appendix. We therefore conclude that the RNA data is reasonably reliable.

The authors published themselves in this very journal work on protein-RNA correlations across cell lines (Edfors, 2016) which should also be cited. The work here extends this work. However, since the 2016 paper was done on highly accurate, targeted RNA and protein measurements, again, it can perhaps be used to QC/VALIDATE the more shotgun/larger-scale quantitation that is done here. Same applies to comparison to Wilhelms 2014 data.

We also included information on this point in the appendix. The Edfors study did not include a targeted RNA quantification approach such as qPCR. Hence, we were unable to do this analysis. The Edfors study analysed 55 proteins across 10 tissues by targeted MS (PRM). Of these, 52 proteins were also identified in our study. Comparison of the copy numbers determined by PRM and our shotgun approach agreed very well. While this is not new, it was reassuring to see that, depending on the tissue, the Spearman correlation coefficients were, on average, at 0.79 and, importantly, the slopes of the regression line were, on average, at 1.05 indicating that the shotgun data did not systematically over- or underestimate copy numbers (see appendix for correlation plots

for all 10 tissues). We also compared protein/mRNA correlations reported by Wilhelm et al. for the 11 tissues that are common between both studies. We found that the data in the current study correlated substantially better for any tissue ranging from 0.42 for Ovary to 0.57 for Adrenal gland (median of 0.52). The respective figures for the Wilhelm study ranged from 0.31 (Thyroid) to 0.56 (Kidney) and had a median of 0.41. Hence, the current data appears to be of higher overall quality than the Wilhelm study. This is not surprising because the current study used directly adjacent cryosections of the same tissue for RNA and protein analysis whereas the Wilhelm study used RNA data deposited in a public repository.

Further, is there ANY data on an assessment of reproducibility of the findings? I did not find it, not for the tonsil, nor the other tissues. At least some assessment of reproducibility of quantitation or identification is crucial for anyone who wants to use this enormous dataset but doesn't know to which extent the quantification is reliable.

This data is indeed available and we apologize for not including it in the first place. The results of this analysis has been added to the appendix. Briefly, for the RNA-Seq data, three samples each of liver and tonsil were analyzed and showed Spearman correlations of between 0.91 and 0.93. For the proteomic data, also three liver and three tonsil data sets were analysed and Spearman correlations of between 0.87 and 0.92 were obtained. This shows that both data types are quite reproducible. The fact that the proteomics data showed slightly lower reproducibility is not unexpected given the much larger quantitative dynamic range of expression compared to mRNA and the much lower coverage by peptides/peptide spectrum matches (PSMs) compared to RNA-Seq (see also further below) which inevitably leads to lower reproducibility particularly for low abundance proteins.

Quantitation of the expressed proteome, tissue-specific and shared proteins.
As the authors state, the extent of brain/testis specific proteins isn't new - but the authors again have the unique chance to validate some of these findings and the QUANTIFY how much of tissue-specific expression is due to our technological inability to detect the proteins (MS not sensitive enough), computational, or biological (i.e. true tissue-specific expression). I.e. for the in-depth tonsil data, how many of the originally tonsil-specific proteins were then detected in the in-depth protein data? That should give an estimate of how many proteins are missed at the basic technology level.

58 of the 73 tonsil-specific (tissue-enriched in our definition) proteins were also in the in-depth data set. The discrepancy arises from the fact that the proteins from the original Trypsin-HCD experiment of all tissues were identified from a single very large search and using the ‘match-between-runs’ option of MaxQuant. This could not be done for the in-depth data that included additional proteases. Even in the in-depth experiment, the LC-MS/MS workflow was overwhelmed by too many peptides to analyse and thus missed some of these proteins. To address the question as to how many proteins are missed at the basic technical level, we note that the in-depth experiment identified 1,112 proteins not covered by the initial Trypsin-HCD workflow that was applied to all tissues. As stated above, the extra proteins span the entire abundance range of the Trypsin-HCD data which either means that the standard trypsin digestion did not provide access to these proteins or that they were simply missed by chance because the tissue was only analysed by Trypsin-HCD once (see above). Of these 1,112 proteins, 608 were also detected in other tissues and are thus not tissue-specific. The remaining 504 proteins were not detected in any other tissue. Of these, 282 were only identified because other proteases were used. But we cannot exclude the possibility that they could have been found in other tissues too if further proteases would have been included in the analysis of the other tissues. Therefore, these proteins are also not necessarily tissue-specific. The remaining 222 proteins from the in-depth tonsil could potentially represent tonsil-specific proteins. GO analysis of these proteins did not uncover any enriched molecular functions or biological processes. The above implies that, indeed a sizeable number of proteins are missed at the basic technological level. We have added information on the overlap/exclusivity of proteins in the standard vs in-depth tonsil analysis to the appendix.

Further, there should be antibodies for some of the new or tissue specific proteins - how many are detected by western blots? Or targeted MS? - The authors don't need to test everything, but it's a shame to miss on the chance to actually quantify the extent of technological limitations and true biological trends.

We have added information on this point to the main manuscript and appendix. Instead of performing western blot analysis, we took advantage of the very extensive antibody based staining data of many tissues in the Human Protein Atlas project. We note here that the IHC staining is, unfortunately, not quantitative (it only comes in 4 categories of high, medium, low, and not detected). Therefore, it is not possible to perform a quantitative comparison between the MS data of the current study and the antibody staining in the HPA project. But we can at least report on whether or not a protein was detected by both methods.

a) Of the 37 validated 'missing' proteins that we identified but that are not (yet) in nextprot, 18 have antibody staining in the current release of the Human Protein Atlas project and all of them show signal in the same tissue they were detected in by MS). This corroborates the detection of these proteins by a different method.

b) Similarly, for 1,270 of the 1,998 tissue specific proteins detected in our study, we found antibody staining in the Human Protein Atlas. In the 29 tissues that are common in HPA and the current study, 775 proteins were detected in the same tissue lending support to the mass spectrometry based data presented here.

c) In addition, we compared our tissue-enriched expression data to the targeted MS (PRM) data acquired by Edfors et al (2016) for 10 human tissues that overlapped with our tissue panel. Incidentally, the Edfors study had data on three tissue-enriched proteins according to our classification. First, the protein MB (myoglobin) was highly tissue-enriched in our data in the heart which was confirmed both by antibody staining and the PRM analysis. Second, the protein PDK1 (3-phosphoinositide-dependent protein kinase-1) was also found to be a heart-enriched protein and the PRM data confirmed this. This protein was detected in all tissues by antibody staining but we note again that the IHC stains are not quantitative. The third example is the protein CANT1 (Soluble calcium-activated nucleotidase 1) which we detected as a prostate-enriched protein. Again, this was confirmed by the PRM measurement but was again detected in most tissues by IHC.

Again, same applies for RNA-level. For the few proteins that did not have detectable RNA levels, why did the authors not do some qPCR to see if the RNAs are truly not there? And if they are there, this would again give us a feel for technical limitations and perhaps, for the first time, a way to quantify these limitations. What I mean - with this large dataset, the authors shouldn't miss out on the chance to provide a baseline dataset on the true tissue-specificity of RNA and protein expression, the true extent of missing RNA or proteins etc.

We have actually removed this part of the manuscript because a re-evaluation of this data uncovered a few technical issues. First, about 600 of these genes were not mapped to the sequence file used for RNA-Seq. Hence, they were never assigned any read mapping. Unfortunately, gene identifier mapping is still an unsolved general issue that is not easily corrected. Of the remaining 227 genes, 133 had mRNA signal but which was below the cutoff of ≥ 1 FPKM that we applied. This left us with 94 genes/proteins for which we had protein but no mRNA evidence. Although these may be genuine, we decided to drop the point for the sake of being conservative because in the context of a total of ~13,500 gene/protein identifications, this number is below our 1% protein FDR.

Detection of alternative splice variants at the protein level.

The authors go into some discussion of protein-level detection of the splice variants. The dominance of one isoform is not new. The authors then say they detect 41% of the variants by the in-depth/targeted proteomics approaches. I would say that this is actually a high fraction (and higher than previous attempts to MS-detect AS variants). I would like to see a more careful connection to existing work, again - rather than just confirming "Oh yes, it's low, we show that too" - I think the authors can move far beyond this and say "Using our in-depth proteomics approach we can indeed identify a large fraction of the splice variants". Somewhat more curiosity but also along the lines of helping researchers know how good non-in-depth proteomics data is: how many splice variants are detected in the 'normal' tonsil data, that hasn't been analyzed in depth?

Regarding the dominance of one isoform: we cited the prior literature in the manuscript but do think that confirming this at the protein level is a significant result. Regarding the figure of 41%, we want to make sure that there is no misunderstanding. We quote this figure for the analysis dealing with single amino acid variants. And the 41% does not relate to all the single amino acid variants (SAAVs) detected by exome sequencing but to the much smaller number of SAAVs identified on the peptide level. Compared to the total number of SAAVs detected by exome sequencing, the number of such peptides is very low (2.4 %) as stated a few lines below in the manuscript. We have

revisited the text to make sure this is properly explained. With respect to the analysis of splice variants: in light of the comment made by the reviewer, we came to realize that we may have indeed downplayed the value of our data for the detection of isoforms too much. We have therefore rephrased the text along the lines suggested and added information to the appendix. As shown in main Figure 4A, the 'normal' tonsil data provided evidence for 4,304 splice variants. In the in-depth analysis, this number increased to about 5,551. Both are indeed encouraging rather than discouraging figures. Still, the same analysis also showed that there were rather few proteins that were detected with more than one isoform in the tonsil tissue (see appendix).

Correlation between and across RNA and protein concentrations.

In my opinion, this is the weakest part of the manuscript and can be shortened. The correlation has been extensively discussed elsewhere (including the authors' own papers) and does not provide new insight. On the contrary, I would argue that some of the presented results are somewhat out of date given other findings.

We agree that the more interesting part of the RNA/protein concentration analysis is in the back-to-back manuscript. Still, we felt that it was important to cover this point at some basic because we foresee that the data we have generated on RNA and protein level will become very useful for researchers trying to better understand the factors governing the control of protein expression in human tissues. Our study provides perhaps the most comprehensive and high quality data set for investigating the relationships between mRNA and protein expression.

At a very basic level, the authors think that much of the scientific community does not realize just how large the differences between RNA and protein expression are and what that implies biologically. As the response below will show again, there is debate in the field about how the various correlations that can be computed can be interpreted in biological terms. We, therefore do indeed hope that our data will, when analyzed in-depth by specialists from all camps, help clarify some of the confusion this reviewer mentions below. We now start this section of the manuscript with this point.

I encourage the authors strongly to incorporate Fortelny's findings (Nature 2017) into their discussions, along with Franks (PlosCB 2017) which is already briefly mentioned. Both papers show that the wider dynamic range of the protein concentrations explains much of what is often discussed as stronger correlations of protein concentrations across tissues compared to RNA concentrations across tissues. There is still a lot of confusion about this inside and outside the field, and with this potentially high-impact publication, the authors shouldn't miss a chance to clarify some of the misconceptions.

The authors agree that there is confusion when it comes to the interpretation of the RNA/protein correlations. What makes the response difficult is that, in our view, neither the Fortelny and Franks papers really helped resolving the confusion or misconceptions despite making interesting observations. The same might be said about our response to the Fortelny paper (PMID: 28748931) or the review published on the topic by the Aebersold lab (Liu et al. Cell, 2016). The authors cannot help thinking that one of the main issues in this context is our collective inability to find common and clear language when talking/writing about correlations within or across tissues or genes, what similar and dissimilar expression means in quantitative terms, what we think are weak correlations, what we mean by dynamic range and so forth. There may not be quite as much actual disagreement but it is remarkable to observe that scientists come to entirely opposing conclusions based on the very same data. We do not think that this point will be resolved as part of this manuscript. Instead, we indeed hope that the part of the community specializing in this topic will use our data to eventually bring clarity. We added a note on this to the revised manuscript.

The Fortelny and Franks papers definitely need to be cited in the interpretation of the correlations line 252. It doesn't invalidate the authors findings, it just requires a shift in thinking that is important to convey to the community. While 'buffering' (discussed in next sentence) might seem more attractive with respect to the biological interpretation, it is important to clarify that much of the observed trends can be explained purely by differences in dynamic range. Ironically, the authors' own figure (figure 2F) nicely illustrates this effect (even though on a totally different sample). Given similar variation and similar numbers of datapoints, the 'squished' distribution on the right automatically has a lower correlation coefficient, DESPITE overall similar variation as the data on the left.

See our comment above. We are citing both papers in the manuscript and have made changes to the text. We show Figure 2E because we think this way of looking at RNA/protein data is actually not very meaningful because it can be misinterpreted and Figure 2F illustrates this by example. On the very example shown, we note that the variance in the data for SYK and EIF4A3 are NOT similar. There is far more variance in the SYK data both for RNA and protein (39-fold for protein; 45-fold for RNA in natural scale) compared to the EIF4A3 data (11-fold for protein; 6-fold for RNA in natural scale). Therefore, while the strong correlation of SYK expression between tissues can be interpreted in terms of its biology, the lack of correlation for EIF4A3 means little if anything. Reviewer #3 appears to side with our interpretation (see below) but again, we acknowledge that differences in opinion/perception exist that we cannot resolve here. Perhaps it would help if, in the future, the analysis of RNA and protein expression would be performed strictly at the level of copies per cell so that the issues regarding measurement units and scales can be eliminated. According to our and other people's experimental data, iBAQ provides a way to do so for large numbers of proteins. We note that the abundance distribution plots for RNA and protein copies we added to the appendix show that the dynamic range of protein expression is much wider than that of RNA expression and that there are orders of magnitude between the copy number distributions of RNA and protein. Hence, the observations we make and the conclusions we draw from the data still appear very plausible to us.

The statement in figure legend for Figure 3A is therefore entirely misleading. It suggests that this finding is new (which it isn't).

We did not mean to imply that the finding is novel. We simply stated: "Global correlation analysis of proteomes and transcriptomes across human tissues. It is apparent that proteomes correlate stronger between tissues than transcriptomes." The authors stand by this statement realizing that this reviewer may not agree (see our point above). In Figure 3A, we are comparing protein quantities across tissues; not the ratio of RNA/protein. And, separately, we compare the RNA quantities across tissues. We then see that the expression of proteins across tissues is more highly correlated than is the case for RNA. If we can assume that both protein and RNA have been measured with similar accuracy, the argument of 'dynamic range' is not very strong. If anything, the RNA data is more accurate than the protein data. Hence, 'buffering' is still an attractive hypothesis for addressing the question how a cell can maintain a certain amount of protein in different cells despite the fact that its underlying mRNA shows differences. We acknowledge that the term 'buffering' does not describe a specific molecular mechanism how this is achieved. In the revised text, we present both the dynamic range and buffering arguments for balance.

(While I find the correlation discussion overall too long, the example in Figure 2F is actually very nice and could be highlighted more. This would happen automatically if the other stuff is shortened.)

Please see our comments above.

uTIS and non-canonical start codons.

This part is very interesting and can be highlighted more. By shortening other parts (see above) this will probably automatically happen.

We agree that this is an interesting part of the data/paper. That said, there are others (see cited references) who have provided much more data/detail on the topic using focused experimentation (mainly employing the TAILS approach) which is why we kept it reasonably short and stress that very high data quality is required for finding genuine such cases.

MINOR

1. It might be a matter of taste, but many paragraphs are a page long or longer. They can easily be split and guide the reader.

Agreed. We have done as suggested.

2. Lines 208 onwards, I do not understand the logic in the second sentence: the dynamic range of transcripts detected by RNA-Seq spanned about four orders of magnitude and that of proteins

detected by mass spectrometry spanned eight orders of magnitude (Fig 2A). This difference alone explains (at least in part) the overall higher coverage of the expressed proteome by RNA-Seq compared to that of LC-MS/MS.

We have clarified this in the manuscript. The logic is as follows: given that there is limited 'sequencing capacity' in both the RNA and protein data, detecting very low abundance molecules will be harder, the wider the dynamic range of expression is. For example, the (paired end) RNA data provided (on average) 18 M reads per tissue. Those 18 M reads are distributed across 4 orders of magnitude of abundance with a bias to the higher abundant transcripts (higher abundance transcripts get more reads). The MS data only provided (on average) ~76,000 peptides and ~284,000 tandem mass spectra (peptide to spectrum matches; PSMs) per tissue and these are distributed over eight orders of magnitude also with a bias for the more abundant proteins. As a result, it is much easier to cover many genes by RNA-Seq than it is to cover the same number by LC-MS/MS.

3. Lines 291 - I find the fact that more protein groups with just 1 member are found in a database restricted to the RNA-seq data (instead of something that includes all variants) redundant and not surprising. The RNA-seq data will likely contain only one or few variants, so not surprising that fewer variants are then found using this as the basis for the proteomics search. This interpretation (unless I am missing something) can be entirely deleted.

While this is indeed not surprising for the reason the reviewer states, the practical consequences are substantial which is why this part of the analysis is not redundant but in fact instrumental. In practice, most proteomic data is searched against Uniprot or similar resources that do not know about variants expressed in a particular tissue under study. Hence, much of the isoform analysis as done here would have been blurred (or mostly not been possible) if only Uniprot would have been used.

4. Line 572 and corresponding figures: if the number of datapoint is low (like here, potentially just 10), correlation coefficients are sensitive to the number of datapoints. Therefore, the authors should probably not mix the discussion of data with 10 vs. all 29 datapoints. At the same overall variability, these different datasets will give different correlation coefficients (think of it this way: 2 points are perfectly correlated, always, you can always fit a straight line through 2 datapoints; three a bit less, four a bit less...).

We agree that that there is great danger in performing correlation analysis on few data points. This is why we required a protein to be observed in a minimum of 10 tissues for the correlation analysis shown in Figure 2E and 2F. We repeated the correlation analysis in Figure 2E for proteins identified in 20 or all 29 tissues and found the same median correlation of 0.36 in all three analysis (see Appendix). As we mentioned above, these correlations should not be over interpreted. Requiring expression in all 29 tissues reduced the number of proteins but did not change the distribution of correlation coefficients. And we also found this distribution to be independent of protein abundance (see appendix).

5. Figure 1B - any way the data has been sorted?

Apologies. The data was shown in reverse alphabetical order of the tissue names. We changed this to an alphabetical order of the tissues. No other sorting was performed.

6. Figure 1C – would be nice to discuss the high-abundance proteins not detected at RNA level, and again to check with qPCR!!!

Please see our response above.

7. Figure 3A - labeling of axes is confusing. And also, either this figure needs to be left out or discussed properly with the proper explanation and citations - the observed correlations are likely ENTIRELY due to the larger dynamic range of the proteomics data.

Please see our response above.

8. Figure 4C's message not entirely clear.

We modified the figure to make the point clearer. Briefly, the grey bars are the number of SAAVs detected by sequence database searching. The blue bars now give the number of synthetic peptides that were successfully synthesized (note that we did not manage to get all of them made for technical reasons) and the orange bars show the number of SAAVs that were confirmed by comparing the experimental tandem mass spectra to the ones of the synthetic peptides. The message of the figure is that many of the SAAV candidates found by simple database searching, as is common in proteomics, are plain incorrect and therefore must be validated by synthetic peptides or some other rigorous means.

Reviewer #3:

The manuscript "A deep proteome and transcriptome abundance atlas of 29 healthy human tissues" by Wang et al reports the generation of RNA sequencing and proteomics data covering the expression of 17,615 transcripts and 13,664 proteins in 29 human tissues. To my knowledge this is the largest, most comprehensive single-study report of this kind. The matched tissue expression data are acquired from adjacent cryosections of the same samples, which allows for a direct comparison of mRNA and protein expression values. Therefore, I have no doubt that the data will be a useful resource to address a range of biological questions regarding gene expression. In addition, the manuscript reports a smaller proteomics dataset optimised for sequence coverage but demonstrates that even such state-of-the-art data are insufficient to power robust proteogenomics analyses on their own, highlighting the need for major technical and computational advances to make proteogenomics a reality.

I have no major concerns with regards to this paper, but several points should be clarified / addressed before considering it:

Sample description: It is unclear how many donors, replicates and actual samples were used. I think this should be described in somewhat more detail.

The information has been added to Table EV1.

Line 121 and others: Is the number in brackets the standard deviation?

Correct, we have clarified this in the manuscript

The missing testis proteome: This is a very interesting observation and I think the authors could elaborate a bit more on these unexpected findings. Are the 300 missing proteins from the HPA project randomly distributed across the RNA expression range or towards the lower end of the range? You mention the latter to be enriched in membrane proteins, possibly explaining a lack of extraction / detection at the protein level. What about the proteins from the other end of the spectrum, those where lack of detection is less likely to be related to their abundance. Are they enriched in any common function, or perhaps secreted? Are they very small?

To clarify, as mentioned in the text and Figure EV1B, these 300 missing proteins have higher than average levels of RNA. GO analysis of these 300 proteins revealed mostly sperm cell specific functions. They are not enriched in small proteins, they are mostly intracellular proteins so extraction should not be an issue and we did not find other obvious reason such as the number of potential tryptic peptides that would explain why these proteins would not be detectable by LC-MS/MS. Therefore, we speculated in the manuscript that there may be reasons other than technical which would explain this behavior but we currently have no evidence for that to be the case.

Tissue enriched expression: The 4.3% tissue-enriched mRNAs contain still a substantial fraction of mRNAs only expressed in sperm but, as you suggest, many of those may not form stable proteins. So, one would expect the number of tissue-specific proteins to be even lower than that, but instead it is higher (5.4%). Could that reflect a sensitivity issue? In the sense that lack of detection of low abundance proteins in some tissues creates the appearance of tissue-specific expression and hence even these low numbers may still be an overestimation of true tissue-specific expression? This should be clarified or discussed.

We are grateful for the comment because it turns out that the stated figures of 4.3% and 5.4% should actually read 0.73% and 0.65%. Apologies for the mistake which has been corrected. As a result, the

difference between the two values is too small to be interpreted. But in general, this reviewer is right in that if we had ultimate sensitivity, the number of tissue-enriched proteins may decrease. This is why our definition of tissue-enriched is: "5-times higher expression than in any other tissue" rather than categorically rejecting proteins that are found in more than one tissue. Hence, even at better sensitivity, most of such proteins would still be marked as tissue-enriched proteins.

Fig 2A: I find this plot misleading because it seems to show FPKM and iBAQs as if they were the same unit (the legend is not so clear). One would intuitively interpret it as showing that proteins are more abundant than mRNAs. While that may be true, one can of course not infer that from comparing unrelated abundance measures.

See also our comment above. We have modified the plot to clarify that protein and RNA are using different units of quantity. We have also added a plot for copy numbers to the appendix. This still shows that there are orders of magnitude in difference between protein and RNA abundance and dynamic range.

Abundance differences: I generally agree with the author's conclusion that there appears to be huge discrepancy between the most abundant mRNAs and proteins in a tissue (Fig 2D). However, the examples shown in Fig 2C raise a few questions. First, it seems that the discrepancy is one-sided in the sense that the most abundant proteins (myosins) also rank very high at the mRNA level. I also noticed that the most abundant mRNA, MT-ATP8, encodes a tiny protein (7 kDa) - could the gene / mRNA / protein size confound the analysis, perhaps through the way FPKMs / iBAQs are calculated? Also, "MT" genes are not just mitochondrial proteins, but they are also encoded by the mitochondrial genome. Could the different processing and location of these mitochondrial mRNAs confound the RNA-seq analysis, i.e. overestimate their abundance relative to nuclear RNAs?

We agree that there is a danger that very high abundant transcript may lead to an overestimation of their abundance on a relative scale. However, the (paired end) RNA-Seq data contained an average of 18 million reads per tissue which should diminish (albeit not entirely prevent) this issue. The fact that the characteristic shown in Figure 2C is different between tissues (see plots for all tissues in the appendix) suggests that this potential bias is not universal and thus reflects the underlying biology rather than technical shortcomings. The example shown in Figure 2C is the heart and here, the discrepancy between RNA or protein levels is particularly high. In the heart, one would expect to find a lot of myosin (which the data shows) as well as a lot of mitochondria owing to the very large need for energy production in this organ. In that sense it is not surprising that the RNA levels for mitochondrial (encoded) proteins are extremely high in this tissue. We do agree that there can be bias in iBAQ values for very small proteins because the number of theoretically detectable peptides is low (we added a note to the manuscript). The RNA-Seq data should not have this bias given the depth of sequencing (see above). We see the same high expression of MT-transcripts in the GTEx database (RNA expression data for human tissues). We can, however, not conclusively say if a bias for MT-transcripts is present or not. One may speculate that MT-transcripts may be shuttled more efficiently into the cytoplasm than 'ordinary' transcripts that are shuttled into the cytoplasm from the nucleus. This may be plausible given that mitochondria are co-localized with the rough ER and so are the ribosomes but we have no proof for such a hypothesis which is why we refrained from adding this to the manuscript.

Fig 2E: It's unclear what this is showing. The text says correlation of mRNA / protein ratios across tissues. I assume that means each correlation coefficient reflects how well mRNA and protein abundances correlate across the 29 tissues, with a median of 0.35. And was that calculated on log₁₀ ratios? The figure legend does not mention mRNAs.

This is correct. We have clarified this in the manuscript.

Fig 2F: This is a very nice example, but maybe the authors should replace the term "similar expression levels" (line 246/247) with a quantitative term, because people have different interpretations of what "similar" means.

Agreed. We added the range of expression to the figure for these two proteins to clarify the meaning of similarity in this context.

Fig. 3B: I'm not sure about the co-inertia analysis (CIA). It is not a commonly used method. If the goal is to show that mRNA and protein abundances are more similar within than across tissues, a simple correlation analysis would probably be sufficient, e.g. colour in the diagonal in Fig. 3A and add the median values. The highlighted tissue groups are a quite selective and don't stand out visually as intuitive clusters. Again, for this purpose maybe PCA or tSNE may be more standard methods better suited for the task. Does CIA add anything else to this analysis? If there is nothing substantial I think the authors should use a more frequently used method to increase comprehensibility.

The main purpose of this plot is to show that the biological information content of RNA and protein expression measurements are broadly similar with respect to tissue identity. The authors think that the CIA plot is a more visual representation than adding correlation values/colors to an already busy Figure 3A. The advantage of using CIA over PCA or tSNE is that both RNA and proteome data can be visualized in a single plot and that the length of the arrows in the plot actually quantifies how far the data sets are apart. This cannot be done using PCA or tSNE. In addition, 29 tissues would also be too small a number for a tSNE plot to show something interesting (it is optimized for visualizing very large numbers of samples). We agree that the tissue clusters are somewhat subjective but can be rationalized by the cell type content of these tissues which provides good reasoning why the proteomes and transcriptomes were more similar to each other than to functionally very different organs. Therefore, we would like to keep the figure.

I could not log in to PRIDE with the details provided.

We apologize for this. We checked the login details and they appear to be fine so we are unable to give a reason for why the data was not accessible to you. We hope it is working now. If not, please let the editor know so that we can look into this in more detail.

In addition to the raw files themselves, the processed data of the paper will be a great resource, but only if they are documented better and made available in more accessible form. For example, table EV1 appears to contain the key datasets. But what exactly does "genes in proteome" mean? Or "transcripts in transcriptome"? Are these FPKMs? Etc... It would also be good to provide the tables as txt or csv files rather than Excel.

The PRIDE submission also contains all the MaxQuant output files in .txt format. The journal required us to send a small readme.txt along with every Excel table. The explanation of the meaning of the different tabs in the Excel file is given in the readme.txt. We have revised the description in the readme.txt files in order to improve clarity of what information is provided.

Line 158: missing space

Corrected.

Reviewer #4:

This manuscript entitled "A deep proteome and transcriptome abundance atlas of 29 healthy human tissues" by Wang et al. provides an expression catalog of proteome and transcriptome from 29 paired healthy human tissues and discusses the expression differences within and across tissues. The authors describe the tissue enriched gene and protein expression and their role in studying the drug targets. Limitations of proteogenomics analyses to identify and validate the tissue specific isoforms and coding variants are also well described. Overall, I feel that this manuscript is well-written and is suitable for publication after the following minor concerns are addressed:

The authors are happy to read that this reviewer liked the work.

1. A description about the correlation of transcriptome from tissues that were also profiled in GTEx project should be provided. This will help assess if the presented differences across tissues were real or partial artifacts of tissue heterogeneity that can be confounded by factors such as sex and age.

We note that we purposefully designed this study to minimize technical artifacts by performing RNA and protein expression profiling on paired samples from directly adjacent tissue sections. This

is a unique feature of this compared to other studies. As requested, we added a comparison of our RNA-Seq data to that of the GTEx database to the appendix. The slopes of the regression lines were very close to unity and the Spearman correlations for all tissues were high ranging from 0.57 to 0.82. We think this is very good overall agreement considering the differences in donors and technical details in acquiring the RNA expression data. The GTEx consortium also analysed tissues of donors of different age and sex and found no confounding influence.

2. This study demonstrates that about 50% of genes have elevated expression in one or more tissues. Because tissue gene expression can also be regulated by isoform switching, it would be informative if the authors could comment on the tissue-specific expression profile at the isoform level.

This is an interesting point but, unfortunately, we were unable to address this globally. This is because one would have to search the entire data (>1,000 MS files) against all RNA-Seq derived protein sequence databases simultaneously in order to avoid mistakes in protein grouping. Unfortunately, neither of the two search engines available to us (Mascot and MaxQuant) support such an analysis and we are not aware of any that would. Figure EV3 shows the number of isoforms detected for all tissues but, unfortunately, we cannot compare the data across tissues without risking many mistakes.

3. The authors should mention how many proteins out of 72 "missing" in neXiProt were identified that meet the HPP guidelines (Deutsch et al. 2016) to qualify for protein level evidence. They could also comment on their expression levels at transcriptome and proteome in tissues and if they are enriched in any tissues.

Please see our comment above. Re-evaluation of the data reduced the figure to 37 proteins because we required a minimum Andromeda score of 100, required detection at the transcript level and required the protein to be validated by comparison of the experimental MS/MS spectrum to that of a synthetic peptide standard. In the original manuscript, the latter two criteria were not applied. Of these 37 proteins 8 qualify for protein level evidence by HPP guidelines (≥ 2 peptides, ≥ 9 amino acids in length). We note that the HPP guidelines use reasonable but *ad hoc* criteria which are likely too conservative and therefore likely miss genuine cases. Comparing spectra of endogenous to synthetic peptides is likely the more objective criterion which is why we added mirror plots of all evaluated cases to PRIDE. The expression levels of the proteins were about a factor 10 below median (iBAQ at log10 scale, 7.4 vs 8.3). Interestingly 15 of these proteins were detected in the fallopian tube, an organ that has not yet been extensively profiled by proteomics. We did not detect any common GO terms for these new proteins. We added some text on this to the manuscript.

4. Are there common genes across tissues that are seen only at transcriptome level but not at the protein level? If yes, what class of processes/functions are they involved in?

There are 579 genes were detected as (mostly low abundant) transcripts in all 29 tissues but we did not find any protein evidence in any of the tissues. GO analysis of these genes only revealed a single term "protein ubiquitination involved in ubiquitin-dependent protein catabolic process" (pvalue=0.0014) and was represented by just 22 genes. Hence, we think this analysis is not conclusive.

5. The authors should show separate plots similar to Figure 3A for all five classes of tissue-specific expression profiles to observe their patterns of transcriptome and proteome expression across the tissues.

We have added these plots to the appendix as requested. The analysis shows that protein levels correlated better between tissues than mRNA levels in all categories.

2nd Editorial Decision

18th December 2018

Thank you for sending us your revised manuscript. We have now heard back from reviewer #2 who was asked to evaluate your study. As you will see below, the reviewer is satisfied with the revised study and thinks that it is suitable for publication, pending a few minor modifications listed in their report below.

REFEREE REPORTS

Reviewer #2:

Wang et al. present a revised manuscript that is much clearer and addresses all confusions and concerns. The new version is easier to read and presents an important census of the human tissue proteome/transcriptome. The authors edited the m/s to soften several strong statements which makes the work more balanced and of higher impact.

The work should be interesting to a wide community of researchers as it provides (and will provide possibly for a longer time) the most complete overview of protein expression in human. It therefore may serve as a gold-standard for many systems-biology analyses that investigate expression regulation across tissues.

Remaining minor comments:

- perhaps worth mentioning the high number of missing/new proteins for testis in the abstract as it's a major finding (in my view)
- line 243 - while it is nicely explained from the numbers, just stating that sampling depth in addition to dynamic range impacts detection of low-abundance molecules might make it even clearer
- I also think the proteogenomics (variants) findings might be good to mention in the abstract

2nd Revision - authors' response

1st January 2019

Reviewer #2:

Wang et al. present a revised manuscript that is much clearer and addresses all confusions and concerns. The new version is easier to read and presents an important census of the human tissue proteome/transcriptome. The authors edited the m/s to soften several strong statements which makes the work more balanced and of higher impact. The work should be interesting to a wide community of researchers as it provides (and will provide possibly for a longer time) the most complete overview of protein expression in human. It therefore may serve as a gold-standard for many systems-biology analyses that investigate expression regulation across tissues.

Remaining minor comments:

- perhaps worth mentioning the high number of missing/new proteins for testis in the abstract as it's a major finding (in my view)*

We have added the finding that hundreds of high abundance mRNAs from testis could not be identified as proteins to the abstract.

- line 243 - while it is nicely explained from the numbers, just stating that sampling depth in addition to dynamic range impacts detection of low-abundance molecules might make it even clearer*

We have added the fact that sampling depth also impacts the ability to detect low abundance molecules.

- I also think the proteogenomics (variants) findings might be good to mention in the abstract*

We have added the specific numbers of single amino acid variants identified by exome sequencing and mass spectrometry to the abstract

Corresponding Author Name: Bernhard Küster

Manuscript Number: MSB-18-8503